# Lipid Nanoparticle Database towards structure-function modeling and data-driven design for nucleic acid delivery

Evan Collins [1,2,3,15], Jungyong Ji [4,15], Sung-Gwang Kim [4,15], Jacob Witten [1,2,5], Seonghoon Kim[4], Richard Zhu [6], Peter Park [7], Minjun Jung [4], Aron Park[4], Rajith S. Manan [2,5], Arnab Rudra [2,5,8], Gyochang Keum[9], Eun-Kyoung Bang [9,10], Jun-O Jin [11], William J. Jeang [2,8,12], Robert Langer [1,2,5,13,14], Daniel G. Anderson [2,5,8,13,14,16] & Wonpil Im [4,7,16]

Lipid nanoparticles (LNPs) are the leading nonviral nucleic acid delivery technology, but LNP structure-function data remains fragmented and non-standardized. Unlike protein engineering which is anchored by the centralized Protein Data Bank, the LNP field lacks a unified repository for systematic analysis. To address this, we develop Lipid Nanoparticle Database (LNPDB) (https://lnpdb.molcube.com), an integrated database and web tool that consolidates structural and functional data for 19,528 LNPs. LNPDB standardizes LNP featurization by encoding lipid composition, experimental methods, and functional results, and generates CHARMM force field files for constituent lipids to enable molecular dynamics simulations. LNPDB also supports future data contributions for continued growth. We examine the utility of LNPDB through two applications: advancing our deep learning model for predicting LNP delivery performance, and simulating bilayer dynamics to identify structural features – bilayer stability and critical packing parameter – that correlate with LNP delivery performance. Altogether, LNPDB provides the digital framework for LNP modeling and data-driven rational design.

Lipid nanoparticles (LNPs) have emerged as the leading nonviral nucleic acid delivery technology across a variety of applications, including genome editing and protein replacement therapies for genetic diseases, and vaccines for infectious diseases and cancer[1]. In recent years, mRNA delivered via LNPs has been essential in combating serious infection and the spread of COVID-19[2]. While LNP delivery systems have demonstrated therapeutic efficacy, the way in which LNP structural composition affects functional delivery of nucleic acids is

[1]Department of Biological Engineering, Massachusetts Institute of Technology, Cambridge, MA, USA. [2]David H. Koch Institute for Integrative Cancer Research, Massachusetts Institute of Technology, Cambridge, MA, USA. [3]Jameel Clinic, Massachusetts Institute of Technology, Cambridge, MA, USA. [4]MolCube Inc., Seoul, Republic of Korea. [5]Department of Chemical Engineering, Massachusetts Institute of Technology, Cambridge, MA, USA. [6]Department of Biology, Massachusetts Institute of Technology, Cambridge, MA, USA. [7]Department of Biological Sciences, Lehigh University, Bethlehem, PA, USA. [8]Department of Anesthesiology, Critical Care and Pain Medicine, Boston Children's Hospital, Boston, MA, USA. [9]Medicinal Materials Research Center, Biomedical Research Division, Korea Institute of Science and Technology, Seoul, Republic of Korea. [10]KHU-KIST Department of Converging Science and Technology, Graduate School, Kyung Hee University, Seoul, Republic of Korea. [11]Department of Microbiology, Brain Korea 21 Project, University of Ulsan College of Medicine, ASAN Medical Center, Seoul, Republic of Korea. [12]Department of Materials Science and Engineering, Massachusetts Institute of Technology, Cambridge, MA, USA. [13]Harvard and MIT Division of Health Science and Technology, Massachusetts Institute of Technology, Cambridge, MA, USA. [14]Institute for Medical Engineering and Science, Massachusetts Institute of Technology, Cambridge, MA, USA. [15]These authors contributed equally: Evan Collins, Jungyong Ji, Sung-Gwang Kim. [16]These authors jointly supervised this work: Daniel G. Anderson, Wonpil Im. ✉e-mail: dgander@mit.edu; wonpil@lehigh.edu

incompletely understood. Greater understanding of the structure-function relationship of LNPs has the potential to facilitate the development of the next-generation of rationally-designed nanomedicines[3].

LNPs for nucleic acid delivery commonly consist of four lipid components[4]. The primary component is the ionizable cationic lipid, which complexes with the negatively charged nucleic acid and facilitates endosomal escape[5]. The other components include the helper lipid, cholesterol, and polyethylene glycol (PEG) lipid[4]. Extensive in vitro and in vivo screening over decades has revealed that varying both the type and ratios of these four lipid components significantly affects LNP delivery performance[5–11]. Yet, the resulting data from these screens have remained dispersed across studies without standardized formatting, limiting systematic analysis.

This challenge of fragmented data in the LNP field differs from the data infrastructure in protein engineering, where the recent success of deep learning models like AlphaFold[12,13] was made possible by the Protein Data Bank (PDB), a centralized repository that compiles over 200,000 protein structures derived from decades of structural biology experiments. The PDB-to-AlphaFold paradigm underscores the foundational role of large, high-quality datasets in enabling deep learning breakthroughs in the biosciences[14]. However, in contrast to protein engineering, the lipid-based nanomedicine field lacks a unified repository for LNP structure-function data, presenting a barrier to machine learning and predictive modeling.

In recent years, there have been efforts to incorporate machine learning into the screening of mRNA LNPs. One prior study used classifier models trained on 584 LNPs with different ionizable lipids to predict delivery efficacy[15]. Another study developed a graph neural network model, AGILE, trained on 1200 LNPs with different ionizable lipids to predict efficacy[16]. Most recently, a group introduced a message-passing neural network architecture, LiON, trained on 8727 LNPs to engineer new best-in-class ionizable lipids[17]. All of these methods, including a recent effort to synchronize data across studies[18], reflect important first steps in bringing machine learning to lipid nanomedicine; however, there are areas for improvement. First, the datasets used to train these models are limited in size and scope, and offer no way to incorporate future LNP data, restricting their long-term utility. Second, these approaches have focused primarily on ionizable lipid design, overlooking the established contributions of helper lipid[9,11], cholesterol[10], and PEG lipid[19,20] compositions and ratios to LNP performance. Third, these studies have relied on representing ionizable lipids as two-dimensional static graphs as input features for model learning, neglecting potentially important three-dimensional conformational and dynamic features. Current experimental techniques like small-angle X-ray scattering (SAXS)[21,22] and cryogenic electron microscopy (cryo-EM)[22,23] are low-throughput, low-resolution, and cost-prohibitive, which makes it challenging to obtain the three-dimensional structural data on lipids needed for modeling.

Towards addressing these limitations to advance data-driven rational design for nucleic acid delivery, here we develop Lipid Nanoparticle Database (LNPDB) (https://lnpdb.molcube.com). LNPDB is an integrated database and web tool that compiles structure-function data for 19,528 LNP formulations, representing 12,845 unique ionizable lipids across 42 publications (as of August 2025). LNPDB standardizes the featurization of LNPs by encoding their lipid composition, experimental methods, and functional results. LNPDB allows users to systematically search and filter the database. Future user contributions are also supported, enabling LNPDB to expand over time as new data are deposited. Additionally, LNPDB provides CHARMM[24] force field topology and parameter files for all constituent lipids, allowing all-atom molecular dynamics (MD) simulations to generate three-dimensional, time-resolved lipid data that can enhance predictive modeling. For rational LNP design, MD simulations offer a new modality to generate dynamic structural data for lipids not readily accessible with current experimental methods.

In this paper, we introduce the curated dataset currently available in LNPDB and outline the functionality of the accompanying web tool. We next examine two applications of LNPDB towards learning structure-function relationships of LNPs. First, we improve our deep learning model LiON for predicting LNP delivery performance. Second, we simulate bilayer dynamics for select LNP formulations and find that two structural features—bilayer stability and critical packing parameter (CPP) of the ionizable lipid—are associated with LNP delivery performance. Recent studies[25–27] in LNP design have used MD to study mRNA LNP behavior, specifically to investigate pH-sensitive structural transitions; however, our study is the first to leverage features extracted from MD simulations to predict LNP performance, providing a physics-based, data-efficient alternative to traditional deep learning models that rely on two-dimensional static chemical structures. Altogether, this work develops LNPDB as a tool to advance LNP modeling and data-driven rational design for nucleic acid delivery.

## Results

### LNPDB is an interactive LNP structure-function data repository

The basis of LNPDB is structure-function data for 19,528 LNP formulations for nucleic acid delivery, representing 12,845 unique ionizable lipids across 42 publications[6,7,11,15–17,28–63]. Additionally, 269 commercially available ionizable lipids are provided (Fig. 1a, Supplementary Fig. 1, and Supplementary Table 1). LNPDB standardizes an encoding strategy for LNPs based on three general classes of features: composition, performance, and simulation (Fig. 1b). Composition features include lipid types and ratios, along with the ionizable lipid-to-nucleic acid ratio. Lipid type is represented by name and SMILES (Simplified Molecular Input Line Entry System[64]) strings, including parsed head-linker-tail substructures for ionizable lipids, as well as separate representations for their +1e protonated states. Experimental features include methods (e.g., mixing, delivery target, route of administration, batching, cargo type, readout technique) and functional results. Simulation features include CHARMM[24] force field topology and parameter files for lipids comprising each LNP formulation, supporting all-atom MD simulations.

We have developed a web tool for LNPDB available at https://lnpdb.molcube.com. The interactive website allows users to view and search the database (Fig. 1c). Users can search for specific LNPs by properties such as library source, atomic characteristics of ionizable lipids, types of helper lipid, cholesterol, and PEG lipid, molar ratio ranges, and experimental properties. Alternatively, users can search for LNPs by ionizable lipid structure or sub-structure, either by selecting from a list of head, linker, and tail groups, or by using a chemical structure drawing sketch tool. The search functionality built into LNPDB allows researchers to systematically analyze the current LNP landscape and identify any underexplored regions of chemical space for potential lipid innovation. Additionally, researchers can deposit their own LNP structure-function data using the standardized template provided on the website. This helps ensure that LNPDB grows over time as the LNP field evolves.

To visualize the diversity of LNPs and ionizable lipids present in LNPDB, representative embeddings were created using UMAP (see "Methods"). The resulting LNP and ionizable lipid landscapes demonstrate clustering patterns that largely correspond to library (i.e., publication) source, suggesting that the individual 42 studies included in LNPDB tend to explore distinct, non-overlapping regions of lipid design space (Fig. 2a). This is reinforced by our finding that within-library LNP pairs exhibit significantly higher similarity than across-library pairs (Supplementary Fig. 2a), and a UMAP of LNP fingerprints from our deep learning model LiON (as discussed below) similarly yields library-specific clusters, albeit with less pronounced separation (Supplementary Fig. 2b).

Beyond global diversity patterns, LNPDB features diverse compositional and experimental features. As shown in Fig. 2b, the

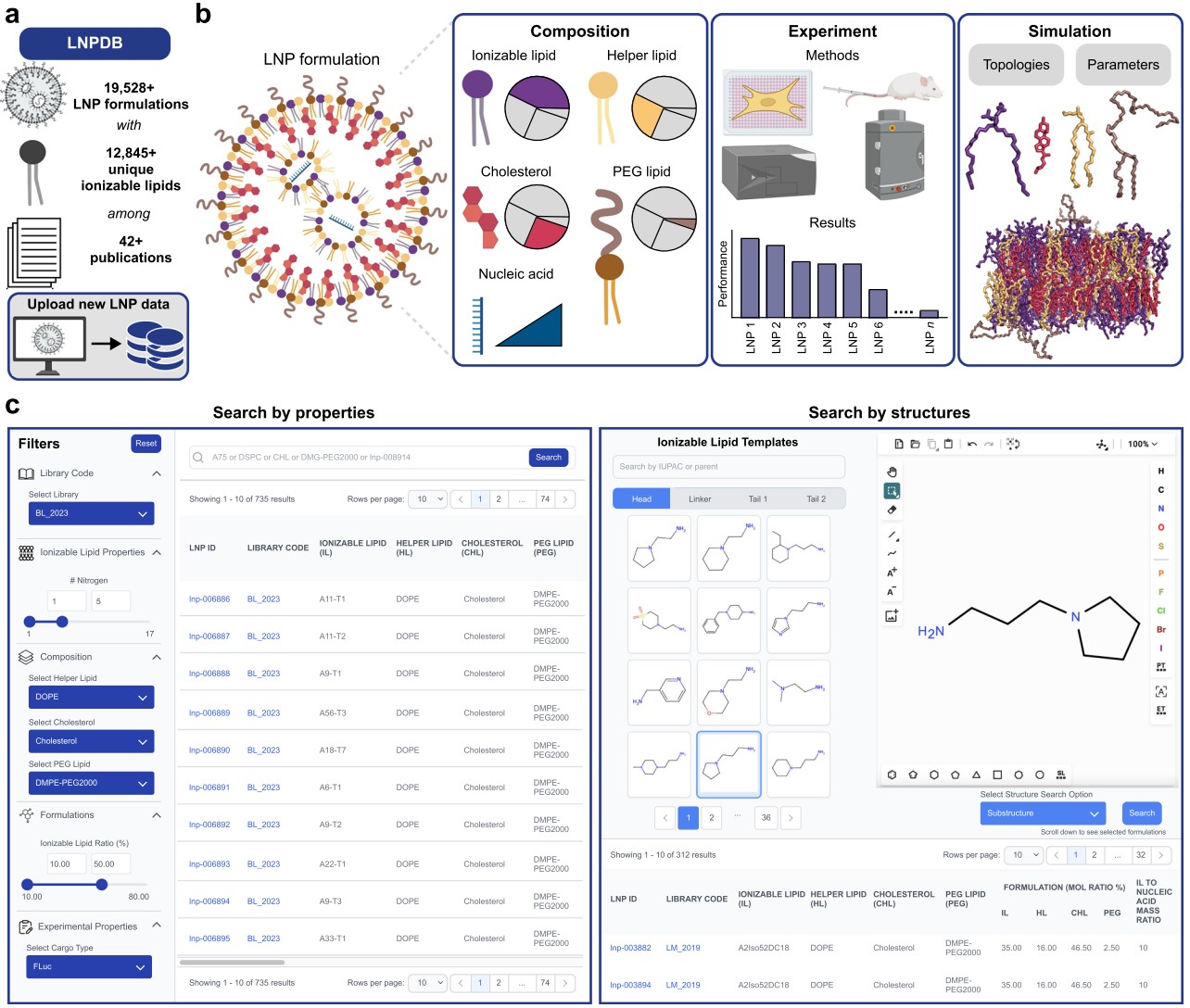

**Fig. 1 | LNPDB is a data repository and web tool for compiling and uploading LNP structure-function data. a** LNPDB compiles structural and functional data for 19,528 LNP formulations for nucleic acid delivery, representing 12,845 unique ionizable lipids across 42 publications and one commercial supplier (as of August 2025). LNPDB is available online (https://lnpdb.molcube.com) to view, search, and upload to the database. **b** Schematic illustration of a single LNP formulation encoded in LNPDB according to three general classes of features: composition (i.e.,

chemical type and ratio), experiment (i.e., methods and results), and simulation (i.e., CHARMM force field topology and parameter files for all-atom MD simulation). Example simulated bilayer for a single LNP formulation (LNP_0003871) composed of its constituent four components at specified ratios. **c** Snapshots from the LNPDB website demonstrating two ways to search the database: by properties (left; e.g., by specific library, composition properties, formulation properties, experimental properties) and by structures (right; e.g., by head, linker, tail, user-drawn structure).

molecular weight of the 12,845 unique ionizable lipids ranges from 201.31 to 3984.45, with a mean of $864.24 \pm 393.12$. The number of nitrogens present in each ionizable lipid ranges from 1 to 17, with a mean of $2.54 \pm 1.41$. There are 12,000 ionizable lipids with only tertiary nitrogen(s); 6437 with both tertiary and secondary nitrogens; 299 with tertiary, secondary, and primary nitrogens; 677 with only secondary nitrogen(s); and 8 with both secondary and primary nitrogens.

As for the other components besides the ionizable lipid, among the 19,528 LNP formulations, the ionizable lipid-to-nucleic acid mass ratio is most often set to 10, with a range from 0.86 to 44.58 and mean of $10.78 \pm 4.45$ (Fig. 2c). The distribution of helper lipid type consists of DOPE (47.7%), DSPC (28.3%), DOTAP (12.6%), none (2.9%), MDOA (2.9%), DDAB (1.8%), 14:0 PA (1.8%), and 18:0 PG (1.8%). The distribution of PEG lipid type consists of DMG-PEG2000 (57.1%), DMPE-PEG2000 (32.4%), unreported (9.2%), none (0.9%), DSG-PEG2000 (0.1%), ALC-0159 (0.1%), DMG-C-PEG2000 (0.1%), C8-Ceramide-PEG2000 (0.1%), C16-Ceramide-PEG2000 (0.1%), DSPE-PEG2000 (0.1%), DMG-PEG5000

(0.01%), DPG-PEG2000 (0.01%), DPG-PEG5000 (0.01%), DSG-PEG5000 (0.01%), DOG-PEG2000 (0.01%), and DOG-PEG5000 (0.01%).

For experimental features, the distribution of nucleic acid cargo consists of mRNA (74.7%), siRNA (19.2%), and pDNA (6.1%) (Fig. 2d and Supplementary Fig. 1b). With respect to the type of cargo encoded by the nucleic acid, the distribution consists of firefly luciferase (90.9%), DNA barcode (3.4%), peptide barcode (2.1%), human erythropoietin (1.4%), Factor VII (0.6%), green fluorescent protein (0.6%), and renilla luciferase (0.3%). The primary delivery target involves in vitro (78.4%), lung epithelium (9.6%), liver (4.6%), muscle (2.5%), spleen (1.3%), multiorgan (1.0%), heart (0.5%), lung (0.5%), and kidney (0.5%). For the preparation method, 93.8% of LNPs were handmixed; the remaining 6.2% were prepared via microfluidics. The readout methods report luminescence (75.2%), discretized luminescence (15.7%), protein abundance (3.9%), cellular uptake (2.5%), editing efficiency (0.7%), LRP6 knockdown (0.6%), diameter (0.5%), zeta potential (0.5%), and percent hemolysis (0.5%). Luminescence measurements quantify LNP

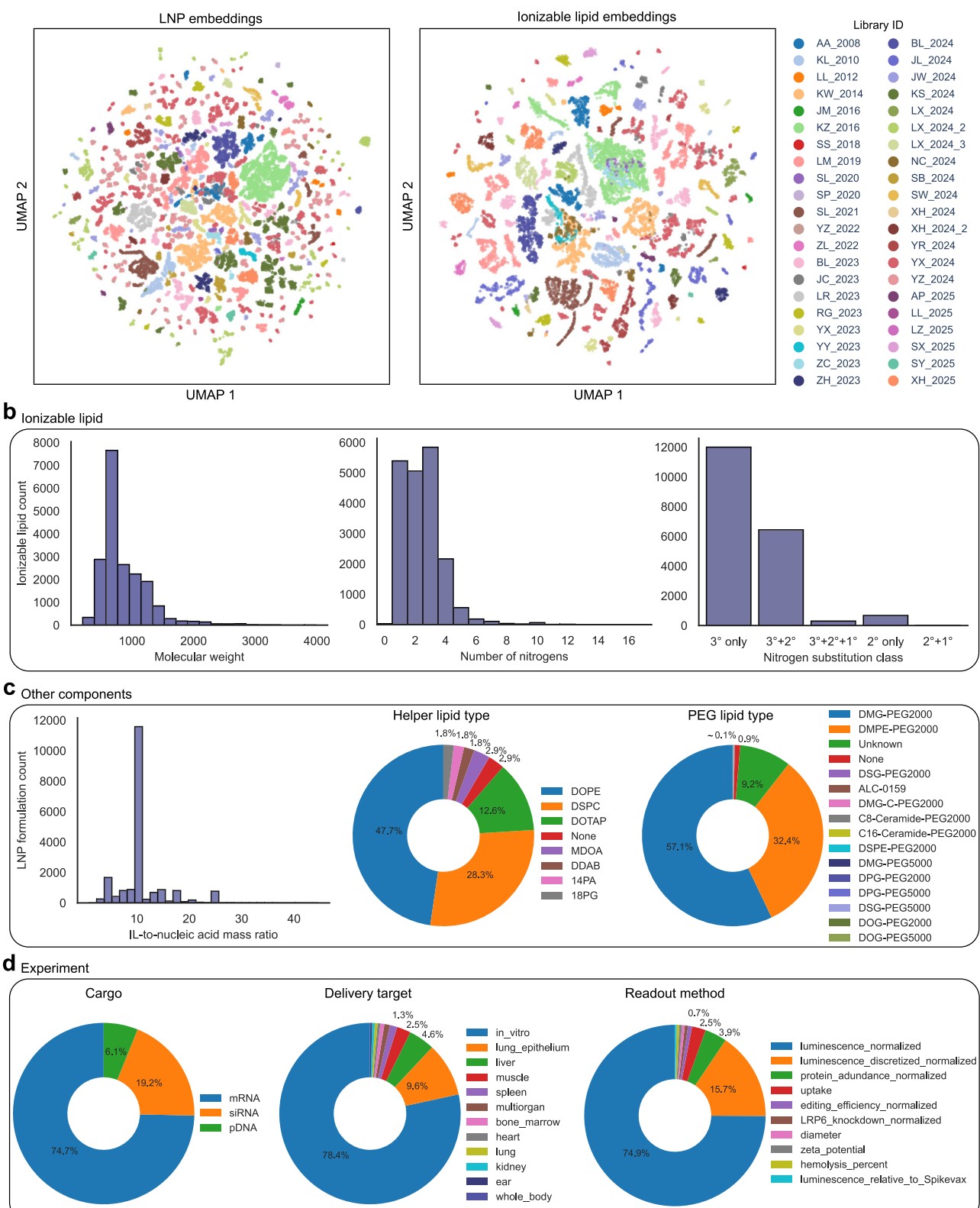

**Fig. 2 | LNPDB includes diverse LNP data from 19,528 formulations across 42 studies. a** UMAP visualization of the high-dimensional embedding landscapes of LNP formulations (left) and unique ionizable lipids (right) compiled in LNPDB, colored according to originating study. **b** Summary statistics of ionizable lipids by molecular weight, number of nitrogens, and nitrogen substitution class.

**c** Summary statistics of ionizable lipid (IL)-to-nucleic acid mass ratio, helper lipid type, and PEG lipid type. **d** Summary statistics of experimental properties by cargo, delivery target, and readout method. Additional summary statistics are shown in Supplementary Fig. 1. Source data are provided as a Source data file.

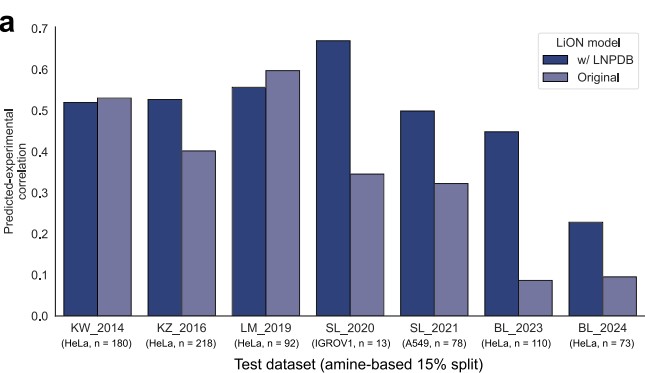

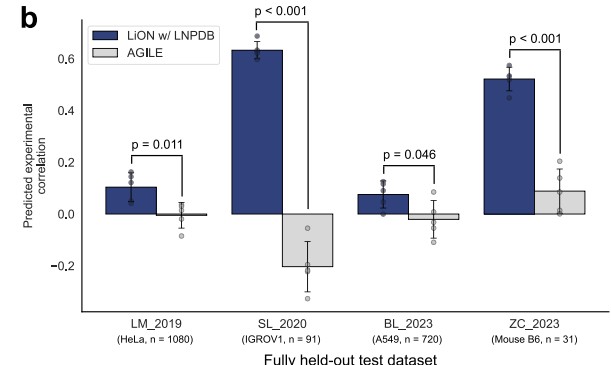

**Fig. 3 | LNPDB facilitates an improved deep learning model for predicting LNP delivery performance. a** Our deep learning model for predicting LNP delivery performance−lipid optimization using neural networks (LiON)−is improved for 5 of the 7 studies evaluated when trained on the LNPDB compared to the original smaller dataset used in our prior study[17]. The plot illustrates the performance of LiON as measured by Spearman correlation between predicted and experimental delivery results on test datasets using an amine-based 70%-15%-15% train-validation-test split. Datasets shared between LNPDB and the original dataset are compared. **b** LiON trained on LNPDB significantly outperforms AGILE[16], an alternative deep learning model for predicting LNP delivery performance, for the 4 fully held-out datasets evaluated. Bars denote mean Spearman correlation coefficient values, and error bars denote ±standard deviation across five train/validation splits with a fixed held-out test set. *p* values resulting from a two-tailed Student's *t*-test are shown. Datasets evaluated are KW_2014[30]; KZ_2016[32]; LM_2019[7]; SL_2020[33]; SL_2021[34]; BL_2023[36]; BL_2024[15]; ZC_2023[42]. Comparable results measured with Pearson correlation are shown in Supplementary Fig. 3. Source data are provided as a Source data file.

delivery performance by reporting the level of nucleic acid transfection in target in vitro or in vivo systems. Additional summary statistics are shown in Supplementary Fig. 1.

## LNPDB facilitates an improved deep learning model for predicting LNP delivery performance

In a prior study[17], we introduced lipid optimization using neural networks (LiON), a deep learning model for learning ionizable lipid design towards predicting LNP delivery efficacy. LiON uses deep message-passing neural networks (D-MPNNs)[65] to learn representations of ionizable lipid structure, while additional formulation details such as component ratios and experimental context are appended as auxiliary features to guide prediction.

The dataset used to train this original version of LiON involved 8727 LNP formulations[17]. Building on this foundation, LNPDB expands the dataset more than twofold by incorporating an additional 10,801 LNP formulations, bringing the total to 19,528. Beyond scale, LNPDB captures a more diverse and descriptive set of features for each formulation. The newly added formulations broaden the diversity of ionizable lipids and also place greater emphasis on varying the types and ratios of the other three LNP components. Moreover, as detailed in the next subsection, unlike the original dataset for LiON, LNPDB includes MD−ready CHARMM force field files for all constituent lipids, introducing a new, physics-based modeling modality altogether for assessing LNP structure-function relationships.

Given that a more robust and diverse dataset can enhance model generalization, we first revisited our deep learning framework LiON to evaluate how training on LNPDB impacts predictive performance compared to the original dataset. To compare model performance, we trained LiON on both the original dataset and LNPDB using a 70−15−15% train-validation-test split, partitioned with respect to amine identity (see "Methods"). Similar to a prior study[17], we evaluated model performance as measured by the correlation between predicted and experimental delivery values. Test datasets shared between the original data and LNPDB were evaluated. The results demonstrate that LiON achieves modestly improved predictive performance for 5 out of the 7 test datasets when trained on the larger LNPDB dataset (Fig. 3a and Supplementary 3a). Overlap between LiON-learned embeddings for the original and LNPDB-added data indicates shared structure-function patterns and densely covered feature space (Supplementary Fig. 4). Moreover, despite limitations of integrating data from multiple

studies as discussed in Methods, LiON models trained across datasets achieved higher predictive performance than those trained on single datasets (Supplementary Fig. 5), suggesting that training across multiple studies in LNPDB enabled LiON to learn more generalizable structure-function relationships.

With our improved LiON model trained on LNPDB, we next sought to compare the predictive performance of our model with another published LNP deep learning model, AGILE[16]. We find that LiON trained on LNPDB achieves significantly better predictive performance compared to AGILE for the 4 held-out test sets evaluated across different delivery targets (Fig. 3b and Supplementary Fig. 3b). Model performance was assessed using five train/validation splits with a fixed held-out test set, with mean correlation coefficients and standard deviations reported across folds. LiON trained on LNPDB has 16-fold more training data compared to AGILE, potentially enabling it to learn a broader range of structure-function relationships and generalize better to unseen data. Altogether, LNPDB supports an improved deep learning model for predicting LNP delivery performance. Importantly, LNPDB establishes a framework of training data for the continued development of next-generation deep learning models for LNP design. Moreover, given that our results demonstrate significant variation in model accuracy across libraries, future research can leverage the training data of LNPDB to design alternative models that may be better suited for the specific LNP design strategy (e.g., helper lipid optimization) or delivery target (e.g., in vivo muscle) under investigation.

## LNPDB facilitates MD simulations to uncover LNP structure-function relationships

By providing CHARMM force field topology and parameter files for each lipid in 19,528 LNP formulations, LNPDB supports all-atom MD simulations of the full dataset, as well as any new formulations constructed from its constituent lipids. LNPDB represents a substantial advancement over existing CHARMM-GUI resources, which were limited to ionizable lipids comprising only 6 different head group types and 5 different tail group types[66]. Moreover, new lipids uploaded to LNPDB will be automatically parametrized for MD simulation as well.

MD offers a complementary alternative to machine learning models for understanding the structure-function relationships of LNPs, differing not only in the type of data produced but also in the way the data is generated. MD simulations yield three-dimensional, temporal structural information for all constituent lipids−not just the

ionizable lipid—capturing both type and ratio. For small organic molecules such as those comprising LNPs, the conformational dynamics and interactions with neighboring molecules may play an outsized role in function, making MD especially informative. The importance of MD is amplified by the limited accessibility of three-dimensional structural data for lipids, as the experimental techniques SAXS[21,22] and cryo-EM[22,23] are low-throughput, low-resolution, and cost-prohibitive. Furthermore, unlike machine learning approaches, MD does not require training data, which is particularly valuable given that, as shown in Fig. 2a, most new ionizable lipids lie outside the distribution of previously characterized structures, potentially limiting machine learning generalization.

To demonstrate how LNPDB can be used to facilitate MD simulations, we simulated the bilayer equilibration process for a subset of LNP formulations and extracted structural features to assess correlation with experimental transfection (Fig. 4a). This use case represents just one of many potential simulation strategies enabled by the database (see "Discussion"). We used the CHARMM force field files of lipids in LNPDB to model representative bilayers for select LNP formulations (Supplementary Table 2). Each leaflet contained approximately 100 lipids. PEG lipids were excluded from our analyses, as they are typically shed prior to endosomal escape[19,67,68], the key bottleneck for effective delivery[69], and the physiological context that we aim to model here.

For a given LNP formulation, two bilayer conditions were simulated: fully-neutral ionizable lipids and half-neutral, half-protonated ionizable lipids. These conditions represent, respectively, the neutral pH prior to cellular uptake and the early endosome environment (pH - 6.5) where roughly 50% of ionizable lipids would be protonated, assuming a pKa of 6.5[70]. All-atom simulations ($N = 134$; 77 fully-neutral, 57 half-protonated) were run using OpenMM[71] for 1.5 μs to allow for bilayer equilibration, which we observe occurring before 1 μs (Supplementary Fig. 6; see "Methods"). Any bilayer that did not remain intact—namely, through ionizable lipids escaping from the membrane—was terminated early. Additional details on simulation conditions are provided in Supplementary Table 2.

Snapshots of the final frame and density profiles of select LNP bilayer systems are shown in Fig. 4b. In all systems, protonated ionizable lipids are generally oriented with their head groups exposed at the membrane—water interface, whereas neutral ionizable lipids exhibited more variable behaviors—some remained at the surface (Fig. 4b center), while others were buried within the hydrophobic core (Fig. 4b left). We observed that certain simulated bilayers from the LM_2019 LNP library[7] were unstable, with ionizable lipids dissociating from the membrane over the course of the simulation (Fig. 4b, right). Notably, we find that simulated bilayer stability is positively associated with experimental transfection, indicating in silico membrane behavior could be a useful screening criterion for the delivery potential of candidate LNPs (Fig. 4c).

Next, we aimed to analyze the simulated bilayers for additional structural features that may correlate with experimental delivery efficacy. One structural feature that we sought to quantify was CPP, which has been used to relate lipid shape to phase behavior and is hypothesized to influence endosomal escape efficiency[22,25,26,69]. We computed CPP values averaged across ionizable lipids for each LNP bilayer according to two different approaches: one based on volume ($CPP_V$) and the other based on radii of gyration ($CPP_{Rg}$) (Fig. 4d; see "Methods"). CPP values < 1 indicate a cone shape (i.e., narrower at the hydrophobic tail region buried in the membrane than at the hydrophilic head group exposed to the aqueous interface). CPP values close to 1 indicate a cylindrical shape. CPP values > 1 indicate an inverted cone shape, which has been implicated in promoting an inverse hexagonal ($H_{II}$) phase, promoting membrane fusion and endosomal escape[22,69]. Our CPP values derived from MD simulations demonstrate comparable relative differences consistent with experimental measurements[72] for two ionizable lipids (Supplementary Fig. 7), with

neutral forms having higher CPP values than their protonated counterparts (Supplementary Figs. 7 and 8), a finding which aligns with prior SAXS experiments[26].

We next analyzed the subset of $N = 34$ different LNP formulations from the LM_2019 library[7] that formed stable bilayers during the simulations to assess if their ionizable lipid CPP values correlate with experimental delivery performance. We find that ionizable lipid CPP significantly predicts LNP performance. For protonated ionizable lipids, the $CPP_V$ approach based on volume yields a Pearson correlation of 0.530 with delivery performance. This association strengthened when stratified by amine group: amine 12 ($r = 0.761$), amine 2 ($r = 0.798$), amine 3 ($r = 0.501$) (Fig. 4e top). The alternative $CPP_{Rg}$ approach based on radii of gyration of the protonated ionizable lipids also shows a robust ($r = 0.546$) correlation with performance, with similarly improved associations when analyzed by amine group: amine 12 ($r = 0.745$), amine 2 ($r = 0.760$), amine 3 ($r = 0.420$) (Fig. 4f top). Compared to protonated ionizable lipids, neutral ionizable lipids in fully-neutral systems demonstrate comparably strong correlations between CPP and delivery performance (Fig. 4e–f bottom). For both CPP approaches, we also find that the significant correlation with performance holds when analyzing neutral ionizable lipids in the half-protonated systems (Supplementary Fig. 9). Moreover, when we focused our analyses on the subset of LNPs with mean CPP values greater than 1—corresponding theoretically to a transition to negative curvature—the correlative performance improved for both the $CPP_V$ method (protonated: overall $r = 0.723$; neutral: overall $r = 0.680$) and $CPP_{Rg}$ method (protonated: overall $r = 0.621$; neutral: overall $r = 0.646$) (Supplementary Fig. 10). Some correlations for amine 3 did not reach statistical significance, likely due to its limited representation of LNPs with CPP > 1. Importantly, overall, these MD-derived correlations with experimental delivery are greater than those of the LiON deep learning model for the LM_2019 fully held-out test set ($r = 0.104$) (Fig. 3b), underscoring the potential of MD as an alternative data-efficient modality for uncovering structure-function relationships. Moreover, we assessed whether the inclusion of PEG lipid or reducing the temperature to 298 K in simulations affected CPP and found no significant effect (Supplementary Fig. 11).

Next, we measured additional structural features of the simulated bilayers: membrane thickness, torque density, and compressibility (see "Methods"). Torque density values of the fully-neutral systems are positively associated with delivery performance ($r = 0.461$); however, high variance in this metric limits the strength of this conclusion. The remaining features for the simulated LNP formulations did not have any significant correlations with performance (Supplementary Fig. 12). Interestingly, $CPP_V$ variance also significantly predicts delivery performance, suggesting that greater ionizable lipid polymorphism could allow for more effective delivery, potentially due to increased capacity to accommodate more inverse-conical lipid geometries (Supplementary Fig. 13).

## Discussion

Despite decades of research and widespread use of LNPs for nucleic acid delivery, no centralized repository exists for compiling LNP structure-function data. Here, we introduce LNPDB, the first large-scale, integrated dataset and web tool for storing, analyzing, and uploading LNP structure-function data.

As of August 2025, LNPDB contains a diverse collection of 19,528 LNP formulations spanning 42 studies and one commercial source, with features capturing lipid chemistry, formulation parameters, experimental conditions, and functional readouts. The web interface allows users to search and filter LNPs by key properties. The database also provides CHARMM force field topology and parameter files for all constituent lipids, facilitating MD simulations on any LNP formulation or custom lipid combination.

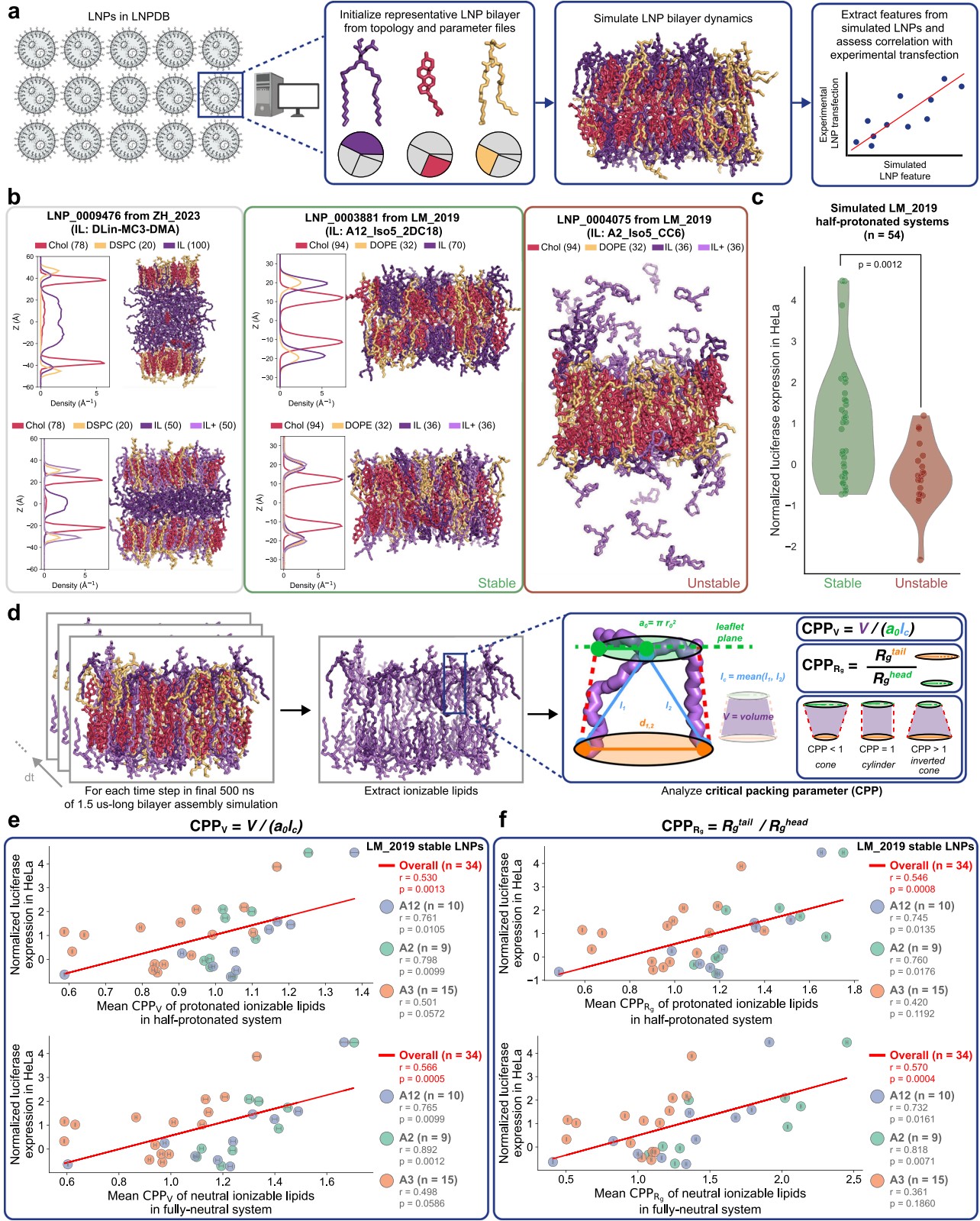

We demonstrate that LNPDB enhances predictive modeling through two distinct yet complementary approaches—machine learning and MD. First, when used to retrain our deep learning model LiON[17], LNPDB doubles the training set size and improves predictive performance across test datasets compared to the original LiON model and the AGILE model[16]. The robust and growing foundation of training data provided by LNPDB can support the future development of even more predictive and generalizable deep learning models for LNP design. Second, we leverage the CHARMM force field files provided in LNPDB to perform MD simulations, uncovering two structural features —bilayer stability and CPP—that correlate with LNP delivery performance for the selected dataset. Notably, CPP values derived from MD

**Fig. 4 | LNPDB facilitates MD simulations of LNP membrane dynamics, uncovering new structure-function relationships towards predicting delivery performance. a** Schematic illustration of the LNP MD simulation workflow enabled by LNPDB: select an LNP formulation of interest, initialize a representative bilayer with the provided CHARMM force field topology and parameter files for the constituent lipids in the selected LNP formulation, run an MD simulation, and extract features from the resulting trajectory to assess correlation with experimental delivery performance. **b** Simulated bilayers for three example LNP formulations available in LNPDB: LNP_0009476 (left; Onpattro formulation from ZH_2023[43]), LNP_0003881 (center; from LM_2019[7]), and LNP_0004075 (right; from LM_2019[7]). Snapshots are taken at 1.5 μs for LNP_0009476 and LNP_0003881, and 100 ns for LNP_0004075. For LNP_0009476 and LNP_0003881, which formed stable bilayers, separate simulation snapshots are shown for the fully-neutral and half-protonated ionizable lipid (IL) conditions; density profiles are provided. For LNP_0004065, which failed to form a stable bilayer as indicated by the escaped ionizable lipids, the simulation snapshot from the half-protonated ionizable lipid condition is shown. **c** Violin plot demonstrating that simulated bilayer stability of $N = 54$ LNP formulations from LM_2019[7] is predictive of experimental delivery performance, as measured in the LM_2019 study by normalized luciferase activity in HeLa cells. $p$ value resulting from

the two-sided Wilcoxon signed-rank test is shown. **d** Method for calculating CPP. Analyses are performed every 1 ns for the final 500 ns of 1.5 μs trajectories of LNP formulations that formed stable, equilibrated bilayers. For each of these timesteps, the coordinates for ionizable lipids are extracted, and the CPP value is computed for each ionizable lipid. **e** For $N = 34$ LNP formulations from LM_2019[7] that formed stable bilayers (subset from original $N = 54$ as $N = 20$ were unstable), the mean $CPP_V$ value—computed as $CPP_V = V/(a_O l_c)$ of ionizable lipids—significantly correlates with experimental delivery performance; this is the case for both protonated ionizable lipids in the half-protonated systems (top), as well as neutral ionizable lipids in the fully-neutral systems (bottom). Linear regression, Pearson $r$ and $p$ values are noted for the overall trend, as well as for each ionizable lipid amine group (A12, A2, A3), which are also represented by point color. **f** Comparable analysis to (**e**) except an alternative method for computing CPP is used: $CPP_{Rg} = R_g^{tail}/R_g^{head}$, where $R_g$ denotes radius of gyration. The $CPP_{Rg}$ method yields similarly robust correlations with experimental delivery performance. Points denote mean values, and error bars denote +/- SEM for $CPP_V$ and standard deviation for $CPP_{Rg}$. Additional details of all bilayer simulations analyzed in this study are provided in Supplementary Table 2. Source data are provided as a Source data file.

show stronger associations with performance than the LiON model's predictions on the same held-out dataset, suggesting that MD provides an orthogonal, data-efficient modality for structure-function discovery.

MD offers unique advantages over deep learning for assessing LNP structure-function relationships. Unlike current machine learning approaches that focus primarily on two-dimensional ionizable lipid structure, MD can inherently account for all four LNP components, as well as their molar ratios, capturing the full multi-component nature of the system. MD simulations generate three-dimensional, time-resolved structural data, providing insights that are inaccessible through static graph-based models and difficult to obtain experimentally via SAXS and cryo-EM. This capability is especially valuable given that the lack of structural definition of nanomedicine remains a major barrier to both therapeutic efficacy and regulatory approval[3]. Furthermore, MD is data-efficient: it does not require large training datasets, making it beneficial for evaluating novel chemistries, as well as underrepresented formulation spaces that preexisting datasets do not effectively capture. This is especially relevant for LNPs, where the large combinatorial space of lipid types and ratios results in sparse data that can pose a major challenge for machine learning model generalization. For example, when screening novel or underrepresented helper lipids, ML models trained on LNPDB may struggle to generalize because the dataset is disproportionately comprised of DOPE, DSPC, and DOTAP (Fig. 2c), reflecting the field's longstanding reliance on these lipids. In such cases, MD can provide complementary value by directly modeling the physical interactions of these underrepresented helper lipids, offering mechanistic insights that are not dependent on prior training data. Moreover, MD can be well-suited for small molecules like lipids, where dynamic shape, orientation, and local interactions can have outsized effects on function. In future applications, MD and machine learning may complement one another, with MD simulations contributing dynamic structural data as input features for deep learning models.

A key limitation of current LNP data—including those compiled in LNPDB—is the difficulty of comparing across studies due to variability in experimental methods (e.g., dose, cell type, animal model, nucleic acid purity, imaging equipment, injection technique, etc.). By establishing LNPDB as a centralized repository, we aim to encourage researchers to incorporate standardized LNP controls in future in vitro and in vivo screens to enable more effective cross-study comparisons. These standardized LNP controls could be Spikevax[2] (50% SM-102, 10% DSPC, 38.5% cholesterol, 1.5% DMG-PEG2000) or Onpattro[73] (50% DLin-MC3-DMA, 10% DSPC, 38.5% cholesterol, 1.5% DMG-PEG2000),

FDA-approved LNPs for COVID-19 and transthyretin-induced amyloidosis, respectively.

Future research should explore experimental validation of MD-derived features such as CPP. The MD bilayer models presented in this work provide a simplified yet informative framework that yields significant correlations with delivery performance for the evaluated dataset. However, future research should leverage the topology and parameter files provided in LNPDB to expand simulation efforts to include additional delivery-relevant phenomena, such as membrane fusion dynamics, interactions with nucleic acids, and dynamic pH sensitivity during endosomal escape. To support MD simulations with larger system sizes, longer time scales, and the inclusion of nucleic acids, we plan to incorporate Martini 3 coarse-grained lipid and nucleic acid parameters[74] in future versions of LNPDB. This will facilitate efficient simulations for many more LNPs to further explore structure-function relationships. Moreover, although this version of LNPDB includes some LNPs with five lipid components, future versions will further incorporate LNPs with more than four components (e.g., additional lipids[8] or lipids conjugated to targeting ligand[75]).

Large, multi-modal datasets will be essential for advancing computational screening approaches in biomolecular design. In protein science, the PDB has provided the foundation for machine learning advances, including the development of AlphaFold[12,13]. In a similar way, LNPDB aims to standardize and centralize structure-function data for LNPs, enabling both machine learning and MD modeling and simulation. Altogether, LNPDB is a tool to advance LNP modeling and data-driven design towards more effective nonviral nucleic acid delivery vehicles.

## Methods
### Data collection
To collect the data for the 19,528 LNPs featured in this initial version of LNPDB, we followed the same method as introduced in our prior study[17]. In summary, publications were selected from the literature based on the presence of large screening datasets, primarily focused on ionizable lipids, to allow for meaningful within-dataset comparisons. Additional publications were selected to broaden the representation of helper lipids, cholesterols, and PEG lipids. SMILES were created for each lipid for each publication. Functional data—most commonly delivery performance—were extracted from published heatmaps and bar plots by digitizing the figures and interpolating values based on either the color scale (heatmaps) or bar height (bar plots) as defined in the accompanying legends. Because delivery values are often reported on different scales across studies and modalities, for each publication and for each delivery context (e.g., in vivo or

in vitro within the same publication), functional delivery data were standardized to have a mean 0 and a standard deviation 1. When raw luminescence values spanned several orders of magnitude, they were first log-transformed prior to standardization. Predictive performance results were separated by publication, as datapoints were treated as directly comparable within individual screens, but not necessarily across different screens or assay modalities. Moreover, standardization was applied to prevent overemphasis of any single dataset. The deep learning models used in this study do train across datasets to maximize the diversity of trainable data, though we recognize that the ability of data from one study to inform structure-function relationships in another is limited by inherent differences in experimental protocols, measurement modalities, and assay sensitivities across studies that may introduce systematic biases. LNPDB introduces experimental condition variables (e.g., solvents, dose) towards bridging studies, but these additions can only partially mitigate the systematic differences across laboratories and experimental setups (see "Discussion").

A total of 269 ionizable lipids from the commercial supplier BroadPharm were also included in LNPDB, representing the full set available on the vendor's website as of June 1, 2024.

## Featurization

LNPDB standardizes an encoding strategy for LNPs based on three general classes of features: composition, performance, and simulation. Additional organizational descriptors include LNP ID, experiment ID, formulation ID, publication link, and publication PubMed ID.

Composition features include ionizable lipid name, ionizable lipid SMILES, ionizable lipid amine name, ionizable lipid amine SMILES, ionizable lipid linker name, ionizable lipid linker SMILES, ionizable lipid tail 1 name, ionizable lipid tail 1 SMILES, ionizable lipid tail 2 name, ionizable lipid tail 2 SMILES, ionizable lipid tail 3 name, ionizable lipid tail 3 SMILES, ionizable lipid tail 4 name, ionizable lipid tail 4 SMILES, ionizable lipid molar ratio, ionizable lipid-to-nucleic acid mass ratio, helper lipid name, helper lipid SMILES, helper lipid molar ratio, cholesterol name, cholesterol SMILES, cholesterol molar ratio, PEG lipid name, PEG lipid SMILES, PEG lipid molar ratio, fifth component lipid name, fifth component lipid SMILES, fifth component molar ratio, aqueous buffer, and dialysis buffer.

Experimental features include mixing preparation method, model (i.e., in vitro or in vivo), model system (e.g., HeLa), model target (e.g., lung), route of administration, cargo, cargo type (i.e., encoded protein), nucleic acid dose, readout method, batching approach, and readout value (i.e., most commonly a measure of delivery performance).

For simulation features, CHARMM topology and parameter files for all lipid components were generated (see *"All-atom molecular dynamics simulations"*). Ionizable lipids were modeled in both neutral and +1e protonated states. A majority of LNPs (13,097) in LNPDB include ionizable lipids that have more than one nitrogen, often with several plausible protonation sites. For simplicity, LNPDB assigns one representative +1e protonated state per ionizable lipid. To select the nitrogen for protonation for each ionizable lipid, the following rule-based decision tree was applied. If the lipid contained only a single nitrogen, that nitrogen was protonated. If multiple nitrogens were present, the nitrogen with the highest priority was protonated based on the following hierarchy: tertiary amine, secondary amine, primary amine, imidazole, pyridine, tertiary aromatic amine, secondary aromatic amine, and primary aromatic amine. Groups comprising amide or sulfonamide structures and quaternary nitrogens were excluded. If multiple candidates of the same class were found, the nitrogen closest to the molecular periphery of the ionizable lipid head—defined as having the greatest graph eccentricity (i.e., the longest existing graph distance to a terminal atom)—was selected. For specific cases within the KZ_2016 dataset involving ionizable lipids with tail amines, the most centrally-located candidate (i.e., the lowest average squared

distance to all other atoms) was chosen. Once selected, the nitrogen was protonated by assigning a +1 formal charge and adjusting the SMILES accordingly. Future research is warranted to explore more accurate, dynamic protonation conditions[76].

## All-atom MD simulations

The CHARMM topology and parameter files for all lipid components were generated as follows: ionizable lipids were modeled in both neutral and +1e protonated states, with parameters assigned manually via CHARMM force field analogy mapping and supplemented by the CGenFF workflow[77]. We used the standard CHARMM force field definitions[78] for helper lipids, cholesterol, and PEG lipids.

Similar to the CHARMM-GUI Membrane Builder that supports diverse lipid types[66,79], LNPDB uses the latest version of the CHARMM C36 additive force field parameters. The CHARMM force field[77] is designed to allow a modular, building-block approach to create force fields for molecules composed of components (blocks) similar to the ones already parametrized. Many topologies and parameters of lipids and carbohydrates in the latest version of the CHARMM C36 force field were generated using this building-block approach; and the generated force fields were further validated by comparing simulations with experimental data. Accordingly, we used a similar building-block approach for the ionizable lipids in LNPDB. We have not yet seen abnormal all-atom simulation behavior (e.g., lipid flip-flop); however, further force field optimization is recommended for specific ionizable lipids if abnormal behavior is observed.

All bilayer systems were assembled using the standalone MolCube Membrane Builder—a commercial software application analogous to the CHARMM-GUI Membrane Builder tool[80]—in membrane-only mode using approximately 100 lipids per leaflet and solvated with TIP3P water[81] and 0.15 M NaCl.

Following the six-step equilibration procedure outlined in the CHARMM-GUI Membrane Builder protocol[80,82], NVT (constant particle number, volume, and temperature) simulations were conducted at 310 K (i.e., temperature of cells treated with LNPs) with strong harmonic positional restraints on lipid heavy atoms and dihedral restraints on ionizable head groups. The restraint force constants were gradually reduced to zero over the six equilibration steps for gradual membrane relaxation. Subsequently, unrestrained NPT (constant particle number, pressure, and temperature) production runs were conducted at 310 K and 1 bar for 1.5 μs using OpenMM with a 4 fs time-step enabled by hydrogen-mass repartitioning (HMR)[83,84]. Temperature was maintained via a Langevin thermostat (collision frequency 1 ps$^{-1}$) and pressure via a semi-isotropic Monte Carlo barostat (coupling interval 0.4 ps)[85,86]. Bonds involving hydrogen were constrained with SHAKE[87]; van der Waals interactions were force-switched off between 10–12 Å[88], and long-range electrostatics were treated by the Particle-Mesh Ewald method with a 12 Å real-space cutoff[89]. For ten bilayer systems, we additionally assessed whether the inclusion of PEG lipid or setting the temperature to 298 K (i.e., temperature of LNP synthesis) in simulations affected CPP and found no significant effect (Supplementary Fig. 11). Information for each simulation system is summarized in Supplementary Table 2. Note that the protonated systems have larger system sizes along the z-axis than their neutral counterparts because the protonated systems require more bulk region to fully solvate the system.

We ran all MD simulations on 48 NVIDIA RTX A5000 GPUs in parallel. Across the neutral systems, the average throughput was 363.86 ± 14.93 ns/day; across the protonated systems, it was 319.11 ± 11.65 ns/day. For the LM_2019 bilayers with a run duration of 1.5 μs, this corresponds to 4.1 days per neutral system and 4.7 days per protonated system. The full LM_2019 simulation batch, run on 48 GPUs (one per simulation), completed in about 10 days. Subsequent CPP calculations for all systems, executed on 384 CPU cores in parallel, finished in 5 h, with comparable compute times for CPP$_V$ and CPP$_{Rg}$.

## Web tool

LNPDB has been developed as a RESTful application built on Django REST Framework and React, with a PostgreSQL backend, together with RDKit for native chemical structure storage and search. Lipid molecules are represented as molecular graphs and indexed via RDKit fingerprints, enabling fast SQL-level structure and substructure queries. User contributions are handled through a CSV upload portal requiring citation metadata. Each submission triggers curator review; data are validated, standardized, and then ingested into PostgreSQL. Approved entries become searchable and visible in all table views and interactive plots via the same API, ensuring seamless integration of new LNP formulations.

## UMAP visualization

UMAP visualizations were created of the high-dimensional embedding landscapes of LNPs and ionizable lipids (Fig. 2a). The embedding landscape for ionizable lipids is represented by the top ten principal components (PCs) of Morgan fingerprints[90] (1024 bits, radius of 3) and the top ten PCs of Mordred descriptors[91]. The embedding landscape for LNPs is represented by the same axes as those for ionizable lipids, plus additional dimensions for molar ratios and the top five Morgan fingerprint PCs and top five Mordred descriptor PCs for helper lipids.

UMAP visualizations were also created of the 300-dimensional embedding landscape (i.e., fingerprints) from the LiON model of LNP formulations (Supplementary Figs. 2b and 4a). Fingerprints were extracted from the penultimate linear layer of the LiON model's feedforward neural network trained on LNPDB.

## Deep learning models

As shown in Fig. 3a, we evaluated the predictive performance of LiON (lipid optimization using neural networks)[17], which is based on the message-passing neural network architecture of *chemprop*[65]. To compare models trained on LNPDB versus the original dataset from our prior study, we computed Spearman correlation values between predicted and experimental delivery outcomes. This evaluation was performed on held-out test sets using a 70–15–15% train-validation-test split by amine, consistent with the approach used in our prior study[17]. Datasets shared between LNPDB and the original dataset are compared.

As shown in Fig. 3b, we evaluated the predictive performance of LiON trained on LNPDB compared to another deep learning model, AGILE[16]. To perform this analysis, each of the four datasets was fully held out as an external test set. For each held-out set, models were trained using five cross-validation folds with 80%-20% train-test splits on the remaining data. Spearman correlation values between predicted and experimental delivery performance for each fully held-out dataset were computed. For LiON trained on LNPDB, holding out an entire dataset reduced the training sample size. In contrast, AGILE maintained its original training size of 1200 LNPs, as the held-out datasets did not overlap with its training data. To run the AGILE model, the GitHub repository provided in the study was referenced, and the HeLa transfection data was used for training[16].

## LNP formulations selected for MD simulations

To assess whether MD simulations could provide meaningful correlations with experimental delivery performance (Fig. 4b–e), we selected $N = 54$ LNP formulations from a prior study (LM_2019) in LNPDB, which introduced an isocyanide-mediated three-component reaction approach for ionizable lipids[7]. For the sake of modeling, we randomly selected to model the subset of LM_2019 LNPs that contain ionizable lipids with amines A12, A2, or A3; isocyanides Iso5 or Iso9; and any alkyl ketone[7]. PEG lipids were excluded from simulations, as they are typically shed prior to endosomal escape[68], the key bottleneck for effective delivery[69], and the physiological context that we aimed to model. This subset, drawn from a single combinatorial ionizable lipid library, was chosen as a representative example of systematic lipid library design commonly employed in the field, while keeping the scope feasible within computational limits.

Additional simulations were conducted for illustrative purposes (Figs. 1b and 4a, b) that contain PEG lipid or the common control ionizable lipids of DLin-MC3-DMA, SM-102, and ALC-0315. Details of all bilayer simulations analyzed in this study are provided in Supplementary Table 2.

## Density profiles

To quantify the spatial distribution of lipid components along the membrane normal (z-axis) as shown in Fig. 4b, density profiles were computed from the final 500 ns of each 1.5 µs MD trajectory. At each frame, atomic coordinates were re-centered to have the membrane center of mass be at $z = 0$. For each lipid molecule, a single representative atom was used to track z-position over time: the hydroxyl oxygen for cholesterol, the phosphorus atom for helper lipids, and the nitrogen atom on the ionizable head group for ionizable lipids. These atom positions were binned along the z-axis to generate one-dimensional density histograms for each lipid type. The profiles were averaged across frames and normalized by bin width to obtain continuous density distributions, reflecting the vertical organization of each component within the bilayer.

## Computing CPP using volume (CPP$_V$)

To quantify ionizable lipid shape, we computed CPP values. For a given stable LNP bilayer simulation, we analyzed each timestep among the final 500 ns of the 1.5 µs trajectory. For each ionizable lipid, atom subsets corresponding to the head group and tail were defined. For the LM_2019[7] LNP bilayers, atoms in the amine and isocyanide groups—both found to be generally positioned at the membrane–water interface—were assigned to the head group; all other atoms (i.e., alkyl ketones) were assigned to the tail group. In line with its conventional formula[25,92], we calculated CPP based on volume as follows:

$$\text{CPP}_V = \frac{V}{a_0 l_c} \quad (1)$$

where $V$ is the volume formed by the ionizable lipid, $a_0$ is the surface area of the head group at the water–membrane interface, and $l_c$ is the average distance between the head group and tail ends (Fig. 4d). The head group area $a_0$ was computed as the cross-sectional area of the circle formed in the membrane plane.

$$a_0 = \pi r_{\text{head}}^2 \quad (2)$$

where $r_{\text{head}}$ is the head group radius, calculated as half the maximum pairwise distance between head atoms in the membrane plane. The tail radius $r_{\text{tail}}$ was computed similarly using terminal tail atoms. The lipid volume $V$ was estimated by modeling the molecule as a truncated cone.

$$V = \frac{1}{3} \pi l_c \left( r_{\text{head}}^2 + r_{\text{head}} r_{\text{tail}} + r_{\text{tail}}^2 \right) \quad (3)$$

These geometrical parameters were computed frame-by-frame for each ionizable lipid molecule across the trajectory to calculate mean CPP$_V$ values. Standard error of the mean (SEM) was also calculated. We observe that CPP$_V$ values exhibit greater variability across lipids and time steps compared to CPP$_{Rg}$. To focus on uncertainty in the central tendency of CPP$_V$ across lipid molecules, SEM was used in place of standard deviation for CPP$_V$ plots.

Lipids with CPP$_V > 1$ exhibit an inverse cone shape, favoring negative curvature, whereas those with CPP$_V < 1$ exhibit a cone shape, favoring positive curvature.

## Computing CPP using radii of gyration (CPP$_{Rg}$)

We also quantified CPP for ionizable lipids using an approach based on radii of gyration, CPP$_{Rg}$. We similarly analyzed the final 500 ns of each 1.5 µs trajectory of stable bilayers. For each ionizable lipid, atom subsets corresponding to the head group and tail were defined, and their centers of mass, $R_{COM}^{head}$ and $R_{COM}^{tail}$ were computed at every frame.

$$R_{COM}^{mid} = \frac{1}{2}\left(R_{COM}^{head} + R_{COM}^{tail}\right) \tag{4}$$

from all atomic coordinates. We then translated each lipid to the origin by subtracting the midpoint. The orientation vector was computed as

$$v = R_{COM}^{head} - R_{COM}^{tail} \tag{5}$$

and aligned to the membrane normal $\hat{z} = [0, 0, 1]$ by rotating the coordinate set through the angle between $v$ and $\hat{z}$. With all lipids consistently oriented, the radius of gyration in the $xy$ plane was computed as

$$R_g = \sqrt{\frac{1}{M}\sum_i m_i(x_i^2 + y_i^2)} \tag{6}$$

where $m_i$ and $(x_i, y_i)$ are the mass and coordinates of atom $i$, and $M = \sum_i m_i$. We recorded the average $R_g$ values for tail and head atoms across all frames—denoted $R_g^{tail}$ and $R_g^{head}$—and computed CPP$_{Rg}$ as

$$CPP_{Rg} = \frac{R_g^{tail}}{R_g^{head}} \tag{7}$$

Lipids with CPP$_{Rg} > 1$ exhibit an inverse cone shape favoring negative curvature, whereas those with CPP$_{Rg} < 1$ exhibit a cone shape favoring positive curvature. Standard deviation was directly computed from the distribution of CPP$_{Rg}$ values over the analysis window (single-block averaging).

## Membrane thickness, torque density, compressibility

The bilayer thickness ($d_B$) is defined as the instantaneous difference between the average $z$-coordinates of phosphate atoms in the upper and lower leaflets:

$$d_B(t) = \langle z_{P,upper}(t)\rangle - \langle z_{P,lower}(t)\rangle \tag{8}$$

We averaged $d_B(t)$ over the final 500 ns of each 1.5 µs trajectory to yield a single representative $d_B$ per system.

Monolayer torque density ($\tau_{mean}$) was calculated from the first moment of the lateral pressure profile $p(z) = p_T(z) - p_N(z)$, where

$$p_T(z) = \frac{p_{xx}(z) + p_{yy}(z)}{2}, \quad p_N(z) = p_{xx}(z) \tag{9}$$

Pressure profiles (0.2 Å bins) were integrated to give leaflet torques:

$$\tau_{upper} = \int_0^{\frac{L_z}{2}} zp(z)dz, \quad \tau_{lower} = \int_{-\frac{L_z}{2}}^0 zp(z)dz \tag{10}$$

and averaged as

$$\tau_{mean} = \frac{\tau_{upper} + \tau_{lower}}{2} \tag{11}$$

Pressure and torque calculations were performed on velocity- and position-recoupled trajectories over the final 500 ns to ensure full equilibration. As context, for a stress-free symmetric bilayer, the monolayer torque density ($\tau$) is related to the bending modulus ($k_c$) and spontaneous curvature ($c_0$) as $\tau = k_c c_0$. A monolayer with positive curvature is convex to the head group side, while negative curvature is concave[66].

The area compressibility modulus $K_A$ was determined from fluctuations in the instantaneous projected bilayer area $A(t)$:

$$K_A = \frac{k_B T \langle A\rangle}{\langle(A(t) - \langle A\rangle)^2\rangle} \tag{12}$$

where $k_B$ is the Boltzmann constant, $T$ the simulation temperature (310 K). Standard deviation was directly calculated from the distribution of $A(t)$ values over the same 500 ns analysis block.

## Reporting summary

Further information on research design is available in the Nature Portfolio Reporting Summary linked to this article.

## Data availability

LNPDB is publicly accessible and can be interactively viewed and downloaded at https://lnpdb.molcube.com/. Source data for Figs. 2–4 and Supplementary Figs. 1–13 are provided with this paper.

## Code availability

Code used to analyze deep learning models and MD trajectories is available on our GitHub repository at https://github.com/evancollins1/LNPDB.

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

## Acknowledgements

This work is supported by Sanofi (D.G.A.), MIT Jameel Clinic (E.C.), NIH grant R33AI161805-05 (D.G.A.), São Paulo Research Foundation Process Number #2024/14345-4 (P.P.), the Nano-Material Technology Development Program through the National Research Foundation of Korea funded by the Ministry of Science and ICT RS-2023-00281553 (J.O.J., W.I.), and the Scale-up TIPS Program RS-2023-00321786 and the TIPA Global R&D Project RS-2025-25458614 funded by the Ministry of SMEs and Startups of Korea (W.I.). Figures created in part using images from BioRender.com.

## Author contributions

E.C., J.J., S.-G.K., J.W., S.K., R.Z., P.P., M.J., A.P., R.S.M., A.R., G.K., E.-K.B., J.-O.J, and W.J.J. created, refined, and analyzed the dataset. E.C., J.J, S.G.K., D.G.A, and W.I. discussed the results and wrote the paper with input from all authors. D.G.A. and W.I. acquired funding. R.L., D.G.A., and W.I. supervised the project.

## Competing interests

D.G.A. receives research funding from Sanofi and is a founder of Orna Therapeutics, Soufflé Therapeutics, and Combined Therapeutics. R.L. is a co-founder and former member of the board of directors of Moderna. R.L. also serves on the board and has equity in Particles for Humanity. For a full list of entities with which R.L. is involved, compensated or uncompensated, see https://www.dropbox.com/scl/fi/ty2b7x8vyebid8ybcbeox/Rev-Langer-COI.pdf?rlkey=lko2srm1qjknm53ck9yns1dfj&e=1&dl=0. W.I. is a co-founder and CEO of

MolCube Inc. The remaining authors have no competing interests to declare.
