## [Transparent Peer Review file · Nature Communications]

Lipid Nanoparticle Database towards structure-function modeling and data-driven design for nucleic acid delivery

Corresponding Author: Dr Daniel Anderson

Version 0:

Reviewer comments:

Reviewer #1

(Remarks to the Author)

This is a timely and important contribution to the lipid nanoparticle (LNP) studies. The construction of the database and integration with molecular dynamics simulations will greatly enhance the ability to design LNPs using a combination of deep learning and physical models. The study is well thought out and will have a major impact on the community. The initial tests already supported the value of the expanded data set (compared to previous work) and the promise of the physically motivated parameters/features. I strongly support the publication of this work in Nat. Commun.

I have only a number of minor questions.

1. For the experimental conditions, it would be interesting to allow the inclusion of additional experimental parameters, such as solution conditions (e.g., buffer).
2. Providing the force field parameters for new lipid formulations would be extremely useful. On the other hand, it is also important to provide some estimate for the confidence of these parameters, so that the users are aware of potential limitations, or are encouraged to further refine the parameters. Pointing this out explicitly will be important, especially to less experienced users.
3. The CPP appears to be less predictive for "amine3". Any possible rationales?
4. How is the torque density related to spontaneous curvature, which is often discussed in the lipid phase/mechanics literature?

(Remarks on code availability)

Data already available at <https://lnpdb.molcube.com>.

Reviewer #2

(Remarks to the Author)

This study employed web-based tools and computational methods to collect structure-function data for 17,116 LNP formulations primarily from 28 publications, integrating them into the LNP database (LNPDDB). The study improves their previously developed deep learning model for predicting LNP delivery efficiency, and also employ molecular modeling tool to examine the relationship between molecular structure and function for 114 LNP formulations using the CHARMM all-atom force field. However, regarding the recommendation for publication of this work, I have several concerns that need to be addressed before proceeding:

1. Data Collection Section: While we acknowledge the authors' focus on the ionizable lipid level, resulting in 10,524 uniquely structured ionizable lipids, and their collection of helper lipids, cholesterol, and PEG lipids from the 28 publications (recognizing their established importance for LNP in vivo activity), the statistical presentation of these auxiliary components in the manuscript appears weakened. Notably, only cholesterol used in classic LNP formulations is specifically mentioned. We are aware that numerous experimental studies have investigated different sterol types (cholesterol, sitosterol, etc.) and PEG lipids (e.g., ALC-0159 used in Covid-19 vaccines). It seems a significant amount of this data is currently missing from the database and requires further incorporation.
2. Data Sharing and Quality: The study mentions that users can download the experimental data they collected. However,

the provided link does not enable a one-click download of the raw data for all LNP formulations. Instead, it only offers an interface for other user groups to upload their experimental data. It is necessary to share all raw data for these LNP formulations, as the quality of data is crucial for model construction. This would significantly advance structure-function research on LNP and has the potential to become as popular as databases like the PDB in the future.

3. Literature Screening Strategy: The literature screening strategy and search keywords should be detailed in the Methods section. The current set of 28 publications clearly does not cover all reported LNP experimental results. Some relevant literature appears to have been excluded, and the reasons for this exclusion need to be clarified.

4. Deep Learning Model Improvement: The authors report significantly improved prediction accuracy for LNP delivery efficiency using their deep learning model, attributed to the expanded dataset (nearly doubled in size). However, the underlying reasons for this improvement are not explained beyond citing the larger number and increased structural diversity of the dataset. A more thorough explanation is needed.

5. Molecular Dynamics (MD) Simulation Selection: The database contains 17,116 LNP formulations, yet MD simulations were performed on only 114 of them. The rationale and criteria for selecting these specific representative formulations are not elaborated upon in the Methods section. The representativeness of this subset for the entire database warrants further discussion.

6. LNP Spherical Model: The study has proposed several models for LNP, with the most prevalent and widely accepted being core-shell structure-like spherical models. The shell layer primarily consists of monolayers, bilayers, or mixtures of both. However, this study only performed self-assembly simulations of the LNP bilayer by removing nucleic acid molecules and PEGylated lipid molecules. This lacks simulation of the complete structure of the LNP formulation. Therefore, the related descriptions and results concerning the LNP formulation require further revision.

7. MD Simulation Temperature: The bilayer structure indeed plays a central role in the proposed LNP structural models, and this study primarily simulates the bilayer assembly process for selected LNP formulations. The Methods section states that simulations were conducted at 310 K. However, this temperature differs from the 298 K typically used for in vitro LNP self-assembly. The justification for using 310 K requires further explanation and the related results require further revision.

8. Critical Packing Parameter (CPP) Correlation: The CPP, previously used to correlate lipid shape with phase behavior, was calculated for ionizable lipids in 34 LNP formulations using both volume and radius of gyration methods in this study. Its correlation with delivery performance yielded intriguing results. For instance, lines 321-339 state: "Interestingly, compared to protonated ionizable lipids, neutral ionizable lipids in fully-neutral systems demonstrate slightly stronger correlations between CPP and delivery performance." The reasons behind this observation deserve further exploration. Furthermore, could the type of nucleic acid cargo (mRNA, siRNA, pDNA) interacting with these lipids influence the CPP values and their correlation with performance?

9. Protonation Site Assignment: In the Methods section (lines 476-490 / 492-496), the authors describe a set of rules used to determine the protonation site(s) for each ionizable lipid, followed by charge assignment for protonation. However, this study seems to have considered only a single protonation site (+1e charge). This approach is clearly suitable for classic ionizable lipids containing only a single tertiary amine (e.g., MC3, SM102). Yet, when an ionizable lipid possesses multiple nitrogen atoms (the study notes the number of N atoms per lipid ranges from 1 to 17), meaning multiple potential protonation sites exist, the applicability and validity of the current method and its results need careful assessment.

(Remarks on code availability)

The authors promise to share all data and code, but we can't find it with the link.

Reviewer #3

(Remarks to the Author)

Summary of the Work:

The manuscript introduces LNPDB, a comprehensive and centralized structure-function database for lipid nanoparticles (LNPs) designed for nucleic acid delivery. The database integrates extensive structural and functional data for 17,116 LNP formulations, standardizing their featurization and providing resources for computational simulations through CHARMM force field files. The authors demonstrate the utility of LNPDB through two applications: (1) enhancement of the LiON deep learning model for predicting LNP performance, and (2) molecular dynamics (MD) simulations that identify structural features correlated with delivery performance. This work significantly advances data-driven approaches in nucleic acid delivery research, providing critical resources analogous to the Protein Data Bank in protein science.

This is a highly impactful, timely, and rigorously conducted study. The authors present a substantial advancement for the field of lipid nanomedicine, addressing a critical gap by providing a standardized, accessible, and expandable platform for systematic analysis and rational design. The integration of molecular dynamics simulations adds valuable dimensionality to the predictive modeling capabilities, providing insights that static structural representations cannot offer.

Revision Suggestions:

1) The authors should explicitly discuss their specific methods for data harmonization and standardization across the diverse experimental conditions included in LNPDB. Given the dataset encompasses varying experimental parameters such as distinct cell lines, diverse target organs, lipid compositions and molar ratios, differing lipid-to-nucleic acid ratios, dosage variations, and varying reporting formats for luminescence measurements (including units and normalization methods), a detailed explanation of how these factors are systematically standardized or normalized is necessary. Clearly articulating this process will enhance understanding of the dataset's consistency, reliability, and suitability for predictive modeling and comparative analyses across different studies.

2) The authors should refer to the article (<https://doi.org/10.1021/acsabm.4c01716>) as this study explicitly investigates the impact of various lipid compositions, including helper lipids, cholesterol, and PEG-lipids, on LNP performance. The

manuscript should clearly delineate how LNPDB advances or expands upon such existing work.

3) The dataset described in the manuscript (as shown in Fig. 2b-c) exhibits significant imbalance. Specifically, despite featuring over 10,000 distinct ionizable lipids, the distribution is heavily skewed: most ionizable lipids contain only up to four nitrogen atoms (despite a range of 1–17 nitrogen atoms), approximately 50% of ionizable lipids have tail lengths concentrated at 18 or 19 carbons (despite a broader range of 3–37 carbons), and most formulations use an ionizable lipid-to-nucleic acid ratio of about 10. Additionally, only seven helper lipid types are represented, with nearly half (45.7%) being DOPE, followed by DSPC (26.9%) and DOTAP (14.4%), leaving minimal representation (<5%) for other types such as MDOA, DDAB, 14:0 PA, and 18:0 PG. The dataset is similarly limited for PEG-lipids, predominantly including only DMG-PEG2000 and DMPE-PEG2000. Given these substantial imbalances, the authors need to thoroughly discuss potential impacts on predictive accuracy, model generalizability, and potential biases introduced by such imbalances. They should explicitly address how these limitations might influence the reliability of data-driven models trained on this dataset and provide a justification or rationale for how the current dataset can still effectively support accurate and generalizable modeling efforts.

4) Although the authors have visually demonstrated diversity using UMAP embeddings (Fig. 2a), the overlaps visible between data points from different sources suggest that additional quantitative measures, such as Tanimoto similarity scores or cluster purity metrics, would be beneficial. Providing such quantitative analyses would strengthen the authors' claims about the distinctiveness of LNP formulations sourced from different studies.

5) The authors have employed Spearman correlation coefficient as their primary evaluation metric (Fig. 3a-b). Given that Spearman correlation assesses rank correlation rather than absolute predictive accuracy, it might not fully capture predictive performance, particularly in practical applications where precise quantitative predictions are essential. The authors should consider or discuss the applicability and advantages of additional metrics such as Pearson correlation, mean absolute error (MAE), root mean squared error (RMSE), or even classification-based metrics if delivery outcomes could be discretized. Justifying the choice of Spearman correlation and/or demonstrating the robustness of their model across these alternative metrics would significantly enhance the manuscript.

(Remarks on code availability)

Version 1:

Reviewer comments:

Reviewer #1

(Remarks to the Author)

The authors have made substantial revision to the ms, and conducted additional analyses (e.g. CPP). I am satisfied with the revisions and support the publication of this timely contribution.

(Remarks on code availability)

Reviewer #2

(Remarks to the Author)

The revised manuscript answered some questions. However, there are still several issues to be addressed before publication.

1. This study presents methodological concerns regarding data integration. The authors employed simple log-transformation and standardization before directly merging data from different studies. However, such basic standardization cannot effectively eliminate systematic biases arising from different laboratories, experimental conditions, and measurement protocols. This approach may create a misleading impression of data comparability and potentially introduce issues in subsequent model training. More rigorous data integration methods should be employed to address data integration challenges.

2. The marked performance variations of the LiON model across different databases are likely attributable to inadequate data integration strategies, which may systematically obscure the true predictive value and distinctive contributions of the LNPDB database. Alternative modeling approaches should be considered, such as dataset-specific modeling or multi-task learning frameworks to more appropriately integrate heterogeneous data sources, thereby yielding more reasonable and reliable model results to reflect the LNPDB quality.

3. The authors used two algorithms to calculate the CPP values of 34 different LNP formulations from the LM_2019 library (lines 340–365) and subsequently correlated them with experimental delivery performance (Fig. 4). The current sample size (34) is too small to represent the 19,528 LNPs and 12,845 unique ionizable lipids in the LNPDB. It is recommended to increase the number of tested formulations to further validate the correlation and conclusion obtained in this study.

4. We have noticed that this study currently considers different quantities of auxiliary lipids (CHOL, DSPC, DOPE) when calculating CPP, with DOPE being the main one. Therefore, whether the differences and changes in the quantity of these helper lipids have an impact on the ionizable lipid CPP should be further compared and explained. The revised manuscript only mentions that the impact of PEG is not significant.

5. The authors noted that when only considering the subset with $CPP > 1$ (Fig. 4 & Fig. S9), the correlative performance improved substantially for both the CPPV and CPPRg methods. Since the subset size was significantly reduced to fewer than 10, the improvement in correlation is expected. However, it remains uncertain whether this conclusion would hold when more ionizable lipids are included.

6. The authors mentioned that using the CPPV method (lines 670-674), "...CPPV values exhibit greater variability across lipids and time steps compared to CPPRg...". Such variability may be normal. It is suggested to perform a cluster analysis to relate the complexity of LNP formulations and the polymorphism of ionizable lipids to delivery efficiency.

7. A comparison of computational equipment and time consumption between the CPPV and CPPRg methods should be provided to help other researchers choose the appropriate method.

(Remarks on code availability)

Reviewer #3

(Remarks to the Author)

I appreciate the authors' careful and thorough revision. In my view, all of my prior concerns have been adequately addressed in both the main manuscript and the Supporting Information. The additions and clarifications substantially improve the clarity, rigor, and reproducibility of the work. I therefore recommend the manuscript for publication.

By way of summary: the revised version clearly documents data processing and standardization, positions the contribution relative to prior efforts, expands and analyzes the dataset to contextualize limitations, and provides quantitative checks that support the principal conclusions. These updates resolve my earlier requests and strengthen the manuscript's impact and transparency.

(Remarks on code availability)

Version 2:

Reviewer comments:

Reviewer #2

(Remarks to the Author)

The authors answered the reviewers' questions.

(Remarks on code availability)

Reviewer #1:

Our Overall Response:

We appreciate the reviewer's valuable feedback. We believe the insightful points have enriched the quality of our manuscript. Shown below are our point-by-point responses. Any new text added to the revised manuscript is shown in red.

Reviewer #1 Point #1

For the experimental conditions, it would be interesting to allow the inclusion of additional experimental parameters, such as solution conditions (e.g., buffer).

Our Response to Reviewer #1 Point #1

We thank the reviewer for motivating us to include additional experimental parameters in LNPDB. We agree features related to solution conditions are useful.

- Accordingly, we have added the new columns *Aqueous_buffer* (e.g., acetate) and *Dialysis_buffer* (e.g., PBS) for all LNPs in LNPDB. In the Featurization subsection of Methods, we hence now also mention "aqueous buffer" and "dialysis buffer" in line 514.
- As for another useful experimental parameter, we have added the column *Dose_ug_nucleicacid* to document the dose of nucleic acid administered in micrograms. In the Featurization subsection of Methods, we hence now also mention "nucleic acid dose" in line 518.
- Moreover, Fig. S1b shown below now includes pie charts depicting diversity of these buffers.

Point-by-point response for consideration at *Nature Communications*

Fig. S1: Additional summary statistics for LNP data in LNPDB. a Number of LNP formulations and unique ionizable lipids for each of the 42 libraries available in LNPDB. There are an additional 269 unique ionizable lipids from BroadPharm (not shown) available in LNPDB. **b** Additional summary statistics of properties by aqueous buffer, dialysis buffer, preparation method, batching method, cargo type, route of administration, and delivery target type. Distributions of experiment value z-scores for each library are plotted. Experiment values not denoting delivery performance were omitted

from this analysis. All boxplots have a box that signifies the interquartile range (IQR; 25th percentile to 75th percentile), a center bar that denotes the median, whiskers that extend up to $1.5 \times \text{IQR}$, and a notch that extends $1.58 \times \text{IQR}/\sqrt{n}$, where n is the sample size for that condition, to estimate the 95% confidence interval. Source data are provided as a Source data file.

- Although not directly related to this point, in response to Reviewer #2 Point #1, we have also significantly expanded LNPDB to better capture the LNP design space, notably expanding diversity in the components besides the ionizable lipid. This has led to marked increases in the overall quantity and diversity of LNPs in LNPDB. The revised database has increased...
 - o from 17,116 to 19,528 LNPs
 - o from 10,524 to 12,845 unique ionizable lipids, and
 - o from 28 to 42 publications
- Two of these new publications include fifth components (i.e., another lipid component besides ionizable, helper, cholesterol, and PEG). Therefore, we have also added new experimental columns to account for fifth components: *fifthcomponent_name*, *fifthcomponent_SMILES*, *fifthcomponent_molratio*.
- Thus, both the number of columns (i.e., features) and rows (i.e., LNPs) have been expanded. All figures are thus updated to reflect this new larger version of LNPDB.

Reviewer #1 Point #2

Providing the force field parameters for new lipid formulations would be extremely useful. On the other hand, it is also important to provide some estimate for the confidence of these parameters, so that the users are aware of potential limitations, or are encouraged to further refine the parameters. Pointing this out explicitly will be important, especially to less experienced users.

Our Response to Reviewer #1 Point #2

We appreciate this suggestion to comment on the confidence of these CHARMM force field parameters. Although this manuscript reflects the first instance of parametrizing thousands of ionizable lipids pertinent to lipid nanoparticle nucleic acid delivery, prior studies provide an important foundation of evaluating the confidence of the CHARMM force field for lipids. We agree it is useful to provide more context and cite these relevant studies.

- Accordingly, in lines 546-555 of Methods, we have added the following sentences, which references important studies for more background about the CHARMM lipid force field. Moreover, in these sentences, we encourage further optimization of parameters as needed (e.g., if inaccurate lipid flip-flop behavior is observed).

“Similar to the CHARMM-GUI Membrane Builder that supports diverse lipid types^{51,62}, LNPDB uses the latest version of the CHARMM C36 additive force field parameters. The CHARMM force field⁶⁰ is designed to allow a modular, building-block approach to create force fields for molecules composed of components (blocks) similar to the ones already parametrized. Many topologies and parameters of lipids and carbohydrates in the latest version of the CHARMM C36 force field were generated using this building-block approach; and the generated force fields were further validated by comparing simulations with experimental data. Accordingly, we used a similar building-block approach for the ionizable lipids in LNPDB. We have not yet seen abnormal all-atom simulation behavior (e.g., lipid flip-flop); however, further force field optimization is recommended for specific ionizable lipids if abnormal behavior is observed.”

Reviewer #1 Point #3

The CPP appears to be less predictive for "amine3". Any possible rationales?

Our Response to Reviewer #1 Point #3

We agree that further investigation of the CPP results is useful; indeed, this comment motivated additional analyses that we believe strengthen this section.

First, rather than assuming the relationship between delivery efficacy and ionizable lipid CPP to be linear across the entire range of CPP values, we also focused in on only the subset of LNPs with mean CPP > 1. This threshold corresponds theoretically to the onset of negative curvature and thus may represent a minimum CPP required to achieve the favorable geometry (with head groups localized to leaflet at aqueous interface) for delivery. As we now demonstrate in Fig. S9 shown below, when we focused our correlation analyses on the subset of LNPs with mean CPP > 1, the correlative performance improved substantially for both the CPP_V and CPP_{Rg} methods. We thank the reviewer for motivating this additional analysis.

Fig. S9: Correlation between CPP of neutral ionizable lipids and experimental delivery performance for the LNPs from LM_2019 which formed stable, equilibrated bilayers with $CPP > 1$. Similar analysis to Figs. 4e-f, S5 but with correlation only computed for LNPs whose ionizable lipids have mean $CPP > 1$, indicating an inverse conical shape conducive to hexagonal phase formation. This $CPP > 1$ thresholding improves correlative performance. CPP values computed using **a** $CPP_V = V / (a_0 l_c)$ and **b** $CPP_{R_g} = R_g^{tail} / R_g^{head}$. Pearson r and p values are noted for the overall trend as well as for each ionizable lipid amine group (A12, A2, A3), which are also represented by point color. Points denote mean values, and error bars denote \pm SEM for CPP_V and standard deviation for CPP_{R_g} . Source data are provided as a Source data file.

- Moreover, whereas amine 3 previously appeared to have lower correlative performance, in this new analysis its correlations are comparable to those of the other two amines. Some of its correlations do remain weaker, however, likely due to the limited number of amine 3 LNPs with $CPP > 1$. Thus, we have added the following sentence in lines 359-360.

“Some correlations for amine 3 did not reach statistical significance, likely due to its limited representation of LNPs with $CPP > 1$.”

Reviewer #1 Point #4

How is the torque density related to spontaneous curvature, which is often discussed in the lipid phase/mechanics literature?

Our Response to Reviewer #1 Point #4

We agree additional clarification on how torque density relates to spontaneous curvature is helpful.

- In lines 731-734, we have added the following sentence. Note that our definition of torque density in this manuscript is the same as that of Park et al., *J. Chem. Inf. Model.* (2021), which we now cite here.

“As context, for a stress-free symmetric bilayer, the monolayer torque density (τ) is related to the bending modulus (k_c) and spontaneous curvature (c_0) as $\tau = k_c c_0$. A monolayer with positive curvature is convex to the head group side, while negative curvature is concave⁵¹.”

Reviewer #2:

Our Overall Response:

We appreciate the review for the thoughtful and constructive feedback. The points provided have motivated revisions which have significantly strengthened the manuscript. Shown below are our point-by-point responses. Any new text added to the revised manuscript is shown in red.

Reviewer #2 Point #1

Data Collection Section: While we acknowledge the authors' focus on the ionizable lipid level, resulting in 10,524 uniquely structured ionizable lipids, and their collection of helper lipids, cholesterol, and PEG lipids from the 28 publications (recognizing their established importance for LNP in vivo activity), the statistical presentation of these auxiliary components in the manuscript appears weakened. Notably, only cholesterol used in classic LNP formulations is specifically mentioned. We are aware that numerous experimental studies have investigated different sterol types (cholesterol, sitosterol, etc.) and PEG lipids (e.g., ALC-0159 used in Covid-19 vaccines). It seems a significant amount of this data is currently missing from the database and requires further incorporation.

Our Response to Reviewer #2 Point #1

We thank the reviewer for this suggestion to expand LNPDB to more fully encompass the field's chemical diversity, especially among the lipid components beyond ionizable lipids.

- Accordingly, we have added new datasets to LNPDB to better capture diversity across all components. Additionally, we have included two publications describing five-component formulations (i.e., those that add a lipid beyond the ionizable, helper, cholesterol, and PEG components), as these are becoming more prevalent in the field. Moreover, following our initial submission, recently published studies evaluating new ionizable lipid libraries have also been added to LNPDB. Altogether, our new version of LNPDB included in this revised manuscript has increased...
 - o from 17,116 to 19,528 LNPs
 - o from 10,524 to 12,845 unique ionizable lipids, and
 - o from 28 to 42 publications
- Here is a more detailed breakdown of the 14 specific publications added to LNPDB. Note that this information is also reflected in the updated Fig. 2, Table S1, and Fig. S1.
 - o Study evaluating from Moderna evaluating SM-102 and 9 other ionizable lipids. This study was added due to the clinical significance of SM-102 for the field.
 - SS_2018 (<https://doi.org/10.1016/j.ymthe.2018.03.010>)

- Study evaluating different cholesterol. This study was added to increase chemical diversity of cholesterol.
 - SP_2020 (<https://www.nature.com/articles/s41467-020-14527-2>)
- Studies evaluating different PEG lipids. These studies were added to increase chemical diversity of PEG lipids.
 - LL_2025 (<https://doi.org/10.1039/D5NR00433K>)
 - LX_2025 (<https://doi.org/10.1016/j.jconrel.2025.01.071>)
- Studies evaluating different fifth component LNP formulations. These studies were added to introduce fifth-component design LNPs into LNPDB.
 - SB_2024 (<https://doi.org/10.1016/j.bioactmat.2024.05.012>)
 - SY_2025 (<https://doi.org/10.1002/adhm.202403366>):
- Studies evaluating new ionizable lipid libraries. These recent studies were added to further increase ionizable lipid diversity.
 - JL_2024
(<https://jnanobiotechnology.biomedcentral.com/articles/10.1186/s12951-024-02919-1>)
 - LX_2024_2 (<https://pubs.acs.org/doi/10.1021/jacs.4c10265>)
 - LX_2024_3 (<https://www.nature.com/articles/s41565-024-01747-6>)
 - SW_2024
(<https://www.sciencedirect.com/science/article/abs/pii/S0168365924007016>)
 - XH_2024_2 (<https://www.nature.com/articles/s41551-024-01267-7>)
 - AP_2025 (<https://www.nature.com/articles/s42004-025-01516-z>)
 - SX_2025 (<https://www.nature.com/articles/s41551-025-01480-y>)
 - XH_2025 (<https://www.biorxiv.org/content/10.1101/2025.02.25.640222v1>)
- Because new datasets have been added, LNPDB now has more rows (i.e., LNP formulations). Additionally, we have added more columns (i.e., features) to LNPDB to better represent each LNP. These new columns include *Aqueous_buffer*, *Dialysis_buffer*, *fifthcomponent_name*, *fifthcomponent_SMILES*, *fifthcomponent_molratio*, *Dose_ug_nucleicacid*.
- The revised Figs. 2, S1 are shown below.

Fig. 2: LNPDB includes diverse LNP data from 19,528 formulations across 42 studies. a UMAP visualizations of the high-dimensional embedding landscapes of LNP formulations (left) and unique ionizable lipids (right) compiled in LNPDB, colored according to originating study. **b** Summary statistics of ionizable lipids by molecular weight, number of nitrogens,

and nitrogen substitution class. **c** Summary statistics of ionizable lipid (IL)-to-nucleic acid mass ratio, helper lipid type, and PEG lipid type. **d** Summary statistics of experimental properties by cargo, delivery target, and readout method. Additional summary statistics are shown in Fig. S1. Source data are provided as a Source data file.

shown) available in LNPDB. **b** Additional summary statistics of properties by aqueous buffer, dialysis buffer, preparation method, batching method, cargo type, route of administration, and delivery target type. Distributions of experiment value z-scores for each library are plotted. Experiment values not denoting delivery performance were omitted from this analysis. All boxplots have a box that signifies the interquartile range (IQR; 25th percentile to 75th percentile), a center bar that denotes the median, whiskers that extend up to $1.5 \times \text{IQR}$, and a notch that extends $1.58 \times \text{IQR}/\sqrt{n}$, where n is the sample size for that condition, to estimate the 95% confidence interval. Source data are provided as a Source data file.

Reviewer #2 Point #2

Data Sharing and Quality: The study mentions that users can download the experimental data they collected. However, the provided link does not enable a one-click download of the raw data for all LNP formulations. Instead, it only offers an interface for other user groups to upload their experimental data. It is necessary to share all raw data for these LNP formulations, as the quality of data is crucial for model construction. This would significantly advance structure-function research on LNP and has the potential to become as popular as databases like the PDB in the future.

Our Response to Reviewer #2 Point #2

We thank the reviewer for this point. Although we had the raw data of the entire LNPDB on our GitHub repo for download, we agree it is helpful to also include on our LNPDB website (<https://lnpdb.molcube.com/>).

- Accordingly, we have added the feature to download the entire LNPDB as a single .csv file on our LNPDB website here: <https://lnpdb.molcube.com/downloads>

Reviewer #2 Point #3

Literature Screening Strategy: The literature screening strategy and search keywords should be detailed in the Methods section. The current set of 28 publications clearly does not cover all reported LNP experimental results. Some relevant literature appears to have been excluded, and the reasons for this exclusion need to be clarified.

Our Response to Reviewer #2 Point #3

We thank the reviewer for recommending additional clarification here. We hope to see further expansion of LNPDB through the community contribution.

- To better clarify our literature screening strategy for the 42 publications in the expanded LNPDB, we have added the following sentences in lines 483-486 of Methods.

“In summary, publications were selected from the literature based on the presence of large screening datasets, primarily focused on ionizable lipids, to allow for meaningful within-dataset comparisons. Additional publications were selected to broaden representation of helper lipids, cholesterol, and PEG lipids.”

Reviewer #2 Point #4

Deep Learning Model Improvement: The authors report significantly improved prediction accuracy for LNP delivery efficiency using their deep learning model, attributed to the expanded dataset (nearly doubled in size). However, the underlying reasons for this improvement are not explained beyond citing the larger number and increased structural diversity of the dataset. A more thorough explanation is needed.

Our Response to Reviewer #2 Point #4

We thank the reviewer for inviting additional explanation here.

- Towards interpreting the impact of including the additional data from LNPDB in our LiON deep learning model, we have now created Fig. S4 shown below. In Fig. S4a, we visualize the LiON-learned LNP fingerprints (i.e., 300-dimensional encodings from the penultimate linear layer of the LiON model's feedforward neural network trained on LNPDB), coloring points by data source (original dataset vs. the data newly included in LNPDB). The two sources are clearly mixed in the embedding space, indicating that new entries populate data neighborhoods already represented by the original set. In Fig. S4b, we further compare cosine-similarity distributions and observe substantial cross-source similarity, consistent with increased local density. Together, these observations suggest that the added data improve coverage of the structure-function manifold, providing a plausible basis for the observed accuracy gains.

Fig. S4: Expanded data of LNPDB blends with original data in LNP embedding space learned by LiON. a Analogous to Fig. S2b but colored by whether the training data were sourced from the original dataset (“Original”) or data newly included in LNPDB (“LNPDB additional”). Note that LNPDB altogether contains the data of both “Original” and

“LNPDB additional”. The LiON fingerprints of these two groups are interspersed, suggesting shared structure–function patterns and that the additional data densifies the representation space. **b** Analogous to Fig. S2a, displaying probability distributions of cosine similarities in LiON embedding space for the top 50 nearest neighbors for each LNP, computed separately within the original dataset and across the original dataset and data newly included in LNPDB. The presence of some high pairwise similarities across the original dataset and the data newly included in LNPDB similarly reinforce the suitability of using the combined dataset to learn shared structure-function relationships. Moreover, LNPs in the expanded dataset that are relatively dissimilar to those of the original dataset broaden the chemical design space, potentially enabling exploration and learning in under-sampled regions. Source data are provided as a Source data file.

- Accordingly, we have added the following sentence in lines 242-244 in Results.
“Overlap between LiON-learned embeddings for the original and LNPDB-added data indicates shared structure–function patterns and a more densely covered feature space (Fig. S4).”
- We have also added the following sentence in lines 596-598 in Methods to explain how the fingerprints were computed.
“UMAP visualizations were also created of the 300-dimensional embedding landscape (i.e., fingerprints) from LiON model of LNP formulations (Figs. S2b, S4a). Fingerprints were extracted from the penultimate linear layer of the LiON model’s feedforward neural network trained on LNPDB.”
- Lastly, although not directly related, this comment overall motivated us to include language more explicitly emphasizing how future research should explore alternative deep learning models tailored to the design strategy or delivery target under investigation. We do not present the machine learning and MD models presented as final, globally optimized solutions; instead, we position LNPDB as the core dataset for developing and benchmarking such models. We hope future research is now equipped to build new machine learning and MD approaches based on the data of LNPDB. Accordingly, we have added the following sentences in lines 254-259 in Results.
“Importantly, LNPDB establishes a framework of training data for the continued development of next-generation deep learning models for LNP design. Moreover, given our results demonstrate significant variation in model accuracy across libraries, future research can leverage the training data of LNPDB to design alternative models that may be better suited for the specific LNP design strategy (e.g., helper lipid optimization) or delivery target (e.g., in vivo muscle) under investigation.”

Molecular Dynamics (MD) Simulation Selection: The database contains 17,116 LNP formulations, yet MD simulations were performed on only 114 of them. The rationale and criteria for selecting these specific representative formulations are not elaborated upon in the Methods section. The representativeness of this subset for the entire database warrants further discussion.

Our Response to Reviewer #2 Point #5

We agree further discussion here is helpful. We now have provided additional reasoning to explain the selection strategy. Note that practical limits of computational time for these all-atom MD simulations were also considered.

- As shown in lines 627-629 of Methods, we now state the following.

“This subset, drawn from a single combinatorial ionizable lipid library, was chosen as a representative example of systematic lipid library design commonly employed in the field, while keeping the scope feasible within computational limits.”

- Moreover, we consider our prior sentences in Discussion emphasizing the use of coarse-grained efforts to improve the scalability of MD simulations are particularly pertinent in this context, as they acknowledge future work should build on this study. Accordingly, we have amplified this section with this additional sentence in lines 465-468, emphasizing how future research could expand these efforts by simulating additional libraries to capture a broader range of LNP design strategies.

“To support MD simulations with larger system sizes, longer time scales, and the inclusion of nucleic acids, we plan to incorporate Martini 3 coarse-grained lipid and nucleic acid parameters⁵⁸ in future versions of LNPDB. This will facilitate efficient simulations for many more LNPs to further explore structure-function relationships.”

Reviewer #2 Point #6

LNP Spherical Model: The study has proposed several models for LNP, with the most prevalent and widely accepted being core-shell structure-like spherical models. The shell layer primarily consists of monolayers, bilayers, or mixtures of both. However, this study only performed self-assembly simulations of the LNP bilayer by removing nucleic acid molecules and PEGylated lipid molecules. This lacks simulation of the complete structure of the LNP formulation. Therefore, the related descriptions and results concerning the LNP formulation require further revision.

Our Response to Reviewer #2 Point #6

We thank the reviewer for inviting additional explanation regarding our modeling choices.

- First, this manuscript is principally intended to advance a digital framework for LNPs that incorporates both machine learning and molecular dynamics (MD). We acknowledge that

these MD bilayer models require simplifying assumptions and are not assumed to represent exhaustive or fully accurate experimental LNP structures. Nevertheless, the results demonstrate that the models can provide meaningful and statistically significant correlations with experimental performance, underscoring their utility despite these simplifications.

- For the MD modeling strategy presented in this manuscript, we decided to model a small portion (i.e., the outermost leaflets of the LNP facing the environments such as solution, plasma membranes, and endosomal membranes) of what the reviewer describes as the core-shell structure-like spherical model. Other MD work too has modeled and analyzed only a portion of LNPs (Tesei et al., *PNAS*, 2024). As mentioned in our response to Reviewer #2 Point #5, practical computational limits prevent full-LNP all-atom MD simulations; it is challenging to model and simulate the entire LNP using all-atom models due to the system size and simulation time. This is why, as we mention in Discussion, our future research will explore coarse-grained models that enable larger-scale simulations. Lastly, when predicting delivery performance *in silico*, smaller-scale models can be advantageous because they enable faster and more efficient simulations. Altogether, we have revised the text of the manuscript at lines 460-471 in Discussion to further clarify these considerations.

“Future research should explore experimental validation of MD-derived features such as CPP. The MD bilayer models presented in this work provide a simplified yet informative framework that provides significant correlations with delivery performance. However, future research should leverage the topology and parameter files provided in LNPDB to expand simulation efforts to include additional delivery-relevant phenomena, such as membrane fusion dynamics, interactions with nucleic acids, and dynamic pH sensitivity during endosomal escape. To support MD simulations with larger system sizes, longer time scales, and the inclusion of nucleic acids, we plan to incorporate Martini 3 coarse-grained lipid and nucleic acid parameters⁷⁵ in future versions of LNPDB. This will facilitate efficient simulations for many more LNPs to further explore structure-function relationships. Moreover, although this version of LNPDB includes some LNPs with five lipid components, future versions will further incorporate LNPs with more than four components (e.g., additional lipids⁸ or lipids conjugated to targeting ligand⁷⁶).”

- Including mRNA in atomistic simulations remains challenging due to computational cost. Previous works have attempted to include only short fragments of mRNA (a few nucleotides) at the coarse-grained level in order to explore global effects and the overall

LNP structure (Tesei et al., *PNAS*, 2024; Grzetic et al., *Mol. Pharm.*, 2024; Kjolbye et al., *ChemRxiv*, 2025). In contrast, our present study focuses on atomistically modeling the outermost leaflets of the LNP, which directly interfaces with external environments such as aqueous solution, plasma membranes, and endosomal membranes. This modeling approach increased our sampling efficiency while retaining atomistic detail, and it ultimately demonstrated significant correlations with delivery performance, which was our central objective. Nevertheless, we agree future research should explore RNA interactions. Accordingly, we have added this text in lines 465-467 in Discussion.

“To support MD simulations with larger system sizes, longer time scales, and the inclusion of nucleic acids, we plan to incorporate Martini 3 coarse-grained lipid and nucleic acid parameters⁷⁵ in future versions of LNPDB.”

- In our initial manuscript, we described our simulations as modeling “the bilayer assembly process”. Because we used Membrane Builder to initialize the bilayer, we have instead replaced “assembly” with “**equilibration**” to be more precise in the revised manuscript.
- With respect to excluding PEG lipids from these models, as we state in Methods in lines 625-629, “**PEG lipids were excluded from our analyses, as they are typically shed prior to endosomal escape^{18,52,53}, the key bottleneck for effective delivery⁵⁴ and the physiological context that we aim to model.**” In other words, we were seeking to model the LNPs in the endosome environment due to its reported importance to delivery performance. PEG lipids are shed before this step; thus, we decided to exclude it. Nevertheless, this reviewer’s comment motivated us to explore whether including PEG lipid would impact our reported CPP results. Thus, we ran 10 additional MD simulations with PEG lipid (whose details are shown in the table below taken from the revised Table S2) to evaluate whether the trend in CPP values differ compared to the comparable simulations without PEG lipid that were already run.

ID	LNP ID	IL name	HL name	# molecules per leaflet					water thickness	XYZ (Å ³)	# Na	# Cl	# waters	# atoms	Duration (ns)	Temp (K)
				IL (P)	IL (N)	HL	Chol	PEG								
LM_1	LNP_0003881	LM_A12Iso52DC18	DOPE	0	35	16	47	2	20	82x82x126	46	46	16849	72269	1,500	310
LM_5	LNP_0003901	LM_A12Iso92DC18	DOPE	0	35	16	47	2	20	83x83x123	46	46	16500	71922	1,500	310
LM_39	LNP_0004885	LM_A12Iso5C11	DOPE	0	35	16	47	2	20	85x85x129	52	52	18825	75549	1,500	310
LM_40	LNP_0004889	LM_A12Iso9C11	DOPE	0	35	16	47	2	20	84x84x112	40	40	14791	64123	1,500	310
LM_41	LNP_0004895	LM_A12Iso5C10	DOPE	0	35	16	47	2	20	84x84x122	46	46	16782	68988	1,500	310
LM_42	LNP_0004899	LM_A12Iso9C10	DOPE	0	35	16	47	2	20	85x85x114	42	42	15548	65978	1,500	310
LM_44	LNP_0004909	LM_A12Iso9C9	DOPE	0	35	16	47	2	20	84x84x141	58	58	20953	81805	1,500	310
LM_46	LNP_0004919	LM_A12Iso9C8	DOPE	0	35	16	47	2	20	83x83x125	48	48	17140	69926	1,500	310
LM_48	LNP_0004929	LM_A12Iso9C7	DOPE	0	35	16	47	2	20	81x81x119	42	42	14941	62897	1,500	310

LM_50	LNP_0004939	LM_A12Iso9CC7	DOPE	0	35	16	47	2	20	84x84x114	42	42	15257	64125	1,500	310
-------	-------------	---------------	------	---	----	----	----	---	----	-----------	----	----	-------	-------	-------	-----

As we show in the new Fig. S10 shown below, including PEG lipids yielded nearly the same trend in CPP values as the comparable simulations without PEG lipids, with $R^2 = 0.981$ for CPP_V and $R^2 = 0.945$ for CPP_{Rg} . We thank the reviewer for motivating this additional analysis. Note that the figure below also shows results for the 10 additional simulations that we ran at 298 K, which we discuss in response to Reviewer #2 Point #7.

Fig. S10: Reducing temperature to 298 K or including PEG lipid in molecular dynamics simulations does not significantly impact CPP trend. To evaluate whether reducing temperature or including PEG lipid affects CPP, 1.5 μ s simulations were run for stable fully-neutral LM_2019 LNPs with amine 12 ($n = 10$), either with a reduced temperature at

298 K or PEG lipid included. **a** Snapshot at 1.5 μ s for LNP_0003881 simulated at 298 K. **b** Snapshot at 1.5 μ s for LNP_0003881 with PEG lipid. PEG denotes C14 lipid with 25 monomeric PEG units. Strong correlations for **c** CPP_v and **d** CPP_{Rg} between systems without PEG at 310 K (as introduced in Fig. 4), systems without PEG at 298 K (shown in blue), and systems with PEG at 310 K (shown in orange). Source data are provided as a Source data file.

Reviewer #2 Point #7

MD Simulation Temperature: The bilayer structure indeed plays a central role in the proposed LNP structural models, and this study primarily simulates the bilayer assembly process for selected LNP formulations. The Methods section states that simulations were conducted at 310 K. However, this temperature differs from the 298 K typically used for in vitro LNP self-assembly. The justification for using 310 K requires further explanation and the related results require further revision.

Our Response to Reviewer #2 Point #7

We thank the reviewer for this suggestion.

- First, we were seeking to model the endosomal environment where the temperature is ~310 K; thus, our MD simulations were carried out at 310 K. We have added these definitions in Methods to clarify in lines 564 and 574-575, respectively.
 “simulations were conducted at 310 K (i.e., temperature of cells treated with LNPs)”
 “setting temperature to 298 K (i.e., temperature of LNP synthesis)”
- To address this point, we assessed the impact of simulating at 298 K (i.e., the temperature at which LNPs are synthesized/self-assembled). Similar to our response to Reviewer #2 Point #6 regarding PEG, we ran 10 additional MD simulations at 298 K (whose details are shown in the table below taken from the revised Table S2) to evaluate whether the trend in CPP values differ compared to the comparable simulations at 310 K that were already run.

ID	LNP ID	IL name	HL name	# molecules per leaflet					water thickness	XYZ (Å ³)	# Na	# Cl	# waters	# atoms	Duration (ns)	Temp (K)
				IL (P)	IL (N)	HL	Chol	PEG								
LM_1	LNP_0003881	LM_A12Iso52DC18	DOPE	0	35	16	47	0	20	83x83x66	12	12	4508	34082	1,500	298
LM_5	LNP_0003901	LM_A12Iso92DC18	DOPE	0	35	16	47	0	20	86x86x67	14	14	4794	35644	1,500	298
LM_39	LNP_0004885	LM_A12Iso5C11	DOPE	0	35	16	47	0	20	83x83x66	12	12	4486	31356	1,500	298
LM_40	LNP_0004889	LM_A12Iso9C11	DOPE	0	35	16	47	0	20	84x84x64	12	12	4310	31528	1,500	298
LM_41	LNP_0004895	LM_A12Iso5C10	DOPE	0	35	16	47	0	20	82x82x64	12	12	4111	29811	1,500	298
LM_42	LNP_0004899	LM_A12Iso9C10	DOPE	0	35	16	47	0	20	81x81x65	12	12	4276	31006	1,500	298
LM_44	LNP_0004909	LM_A12Iso9C9	DOPE	0	35	16	47	0	20	83x83x65	12	12	4317	30709	1,500	298
LM_46	LNP_0004919	LM_A12Iso9C8	DOPE	0	35	16	47	0	20	82x82x65	12	12	4386	30496	1,500	298
LM_48	LNP_0004929	LM_A12Iso9C7	DOPE	0	35	16	47	0	20	83x83x64	12	12	4178	29452	1,500	298
LM_50	LNP_0004939	LM_A12Iso9CC7	DOPE	0	35	16	47	0	20	84x84x63	12	12	4530	30788	1,500	298

As we shown below in the new Fig. S10, changing the temperature to 298 K yielded nearly the same trend in CPP values as the comparable simulations at 310 K, with $R^2 = 0.981$ for CPP_V and $R^2 = 0.986$ for CPP_{Rg} . Note that the figure below also shows results for the 10 additional simulations that we ran with PEG lipid, which we discussed in response to Reviewer #2 Point #6. We also find insignificant difference in CPP trends between the simulations at 298 K and the simulations with PEG lipid at 310 K. We thank the reviewer for motivating this additional analysis. Accordingly, we add this sentence in lines 363-365 in Results and in lines 573-575 in Methods.

“Moreover, we assessed whether the inclusion of PEG lipid or reducing temperature to 298 K in simulations affected CPP and found no significant effect (**Fig. S10**).”

“For ten bilayer systems, we additionally assessed whether the inclusion of PEG lipid or setting temperature to 298 K (i.e., temperature of LNP synthesis) in simulations affected CPP and found no significant effect (**Fig. S10**).”

Fig. S5: Reducing temperature to 298 K or including PEG lipid in molecular dynamics simulations does not significantly impact CPP trend. To evaluate whether reducing temperature or including PEG lipid affects CPP, 1.5 μ s-long simulations were run for stable fully-neutral LM_2019 LNPs with amine 12 ($n = 10$), either with a reduced

temperature at 298 K or PEG lipid included. **a** Snapshot at 1.5 μ s timepoint for LNP_0003881 simulated at 298 K. **b** Snapshot at 1.5 μ s timepoint for LNP_0003881 with PEG lipid. PEG denotes C14 lipid with 25 monomeric PEG units. Strong correlations for **c** CPP_V and **d** CPP_{Rg} between systems without PEG at 310 K (as introduced in Fig. 4), systems without PEG at 298 K (shown in blue), and systems with PEG at 310 K (shown in orange). Source data are provided as a Source data file.

Reviewer #2 Point #8

Critical Packing Parameter (CPP) Correlation: The CPP, previously used to correlate lipid shape with phase behavior, was calculated for ionizable lipids in 34 LNP formulations using both volume and radius of gyration methods in this study. Its correlation with delivery performance yielded intriguing results. For instance, lines 321-339 state: "Interestingly, compared to protonated ionizable lipids, neutral ionizable lipids in fully-neutral systems demonstrate slightly stronger correlations between CPP and delivery performance." The reasons behind this observation deserve further exploration. Furthermore, could the type of nucleic acid cargo (mRNA, siRNA, pDNA) interacting with these lipids influence the CPP values and their correlation with performance?

Our Response to Reviewer #2 Point #8

We thank the reviewer for this suggestion and agree further exploration into our CPP results would strengthen the manuscript.

- Rather than assuming the relationship between delivery efficacy and ionizable lipid CPP to be linear across the entire range of CPP values, we also focused in on only the subset of LNPs with mean CPP > 1. This threshold corresponds theoretically to the onset of negative curvature and thus may represent a minimum CPP required to achieve the favorable geometry (with head groups localized to leaflet at aqueous interface) for delivery. As we now demonstrate in Fig. S9 shown below, when we focused our correlative analyses on the subset of LNPs with mean CPP > 1, the correlative performance improved substantially for both the CPP_V and CPP_{Rg} methods. We thank the reviewer for motivating this additional analysis.

Fig. S9: Correlation between CPP of neutral ionizable lipids and experimental delivery performance for the LNPs from LM_2019 which formed stable, equilibrated bilayers with $CPP > 1$. Similar analysis to Figs. 4e-f, S5 but with correlation only computed for LNPs whose ionizable lipids have mean $CPP > 1$, indicating an inverse conical shape conducive to hexagonal phase formation. This $CPP > 1$ thresholding improves correlative performance. CPP values computed using **a** $CPP_V = V / (a_0 l_c)$ and **b** $CPP_{R_g} = R_g^{tail} / R_g^{head}$. Pearson r and p values are noted for the overall trend as well as for each ionizable lipid amine group (A12, A2, A3), which are also represented by point color. Points denote mean values, and error bars denote +/- SEM for CPP_V and standard deviation for CPP_{R_g} . Source data are provided as a Source data file.

- Whereas neutral ionizable lipids have slightly lower correlative performance in the original correlation analysis, in this new analysis the protonated ionizable lipids have slightly higher correlative performance. However, the observed differences are small and may fall within the variability expected for this type of analysis. Thus, we have revised the manuscript in 349-350 to highlight that protonated and neutral ionizable lipids perform comparably, rather than emphasizing differences.

“Compared to protonated ionizable lipids, neutral ionizable lipids in fully-neutral systems demonstrate comparably strong correlations between CPP and delivery performance.”
- As noted in our response to Reviewer #2 Point #6, it is challenging to include nucleic acid cargo in our current all-atom simulations, and further studies are warranted to examine the influence of nucleic acids on CPP.

Reviewer #2 Point #9

Protonation Site Assignment: In the Methods section (lines 476-490 / 492-496), the authors describe a set of rules used to determine the protonation site(s) for each ionizable lipid, followed by charge assignment for protonation. However, this study seems to have considered only a single protonation site (+1e charge). This approach is clearly suitable for classic ionizable lipids containing only a single tertiary amine (e.g., MC3, SM102). Yet, when an ionizable lipid possesses multiple nitrogen atoms (the study notes the number of N atoms per lipid ranges from 1 to 17), meaning multiple potential protonation sites exist, the applicability and validity of the current method and its results need careful assessment.

Our Response to Reviewer #2 Point #9

We thank the reviewer for this feedback. We have revised the manuscript as described below.

- In lines 525-536, we include additional details regarding the decision tree that was applied to select the single protonation site for each ionizable lipid.
“To select the nitrogen for protonation for each ionizable lipid, the following rule-based decision tree was applied. If the lipid contained only a single nitrogen, that nitrogen was protonated. If multiple nitrogens were present, the nitrogen with the highest priority was protonated based on the following hierarchy: tertiary amine, secondary amine, primary amine, imidazole, pyridine, tertiary aromatic amine, secondary aromatic amine, and primary aromatic amine. Groups comprising amide or sulfonamide structures and quaternary nitrogens were excluded. If multiple candidates of the same class were found, the nitrogen closest to the molecular periphery of the ionizable lipid head – defined as having the greatest graph eccentricity (i.e. the longest existing graph distance to a terminal atom) – was selected. For specific cases within the KZ_2016 dataset involving ionizable lipids with tail amines, the most centrally-located candidate (i.e., the lowest average squared distance to all other atoms) was chosen.”
- We have opted to build LNPDB so each ionizable lipid has the neutral form and one plausible protonated form. Due to practical considerations regarding storage (as the number of possible protonation forms goes up to 2^{17} for the ionizable lipids with 17 nitrogens) and our view that the more robust solution lies in improved pKa measurement/prediction and dynamic protonation conditions, we have opted to maintain our inclusion of one plausible protonated form. We have included these sentences in lines 537-538 to address this perspective.
“Future research is warranted to explore more accurate, dynamic protonation conditions⁶¹.”

To further emphasize this future research direction of dynamic pH simulations, we now include a citation in the above sentence for Jansen et al., *J. Chem. Inf. Model.* (2024), which describes a software tool to allow dynamic protonation states in molecular dynamics simulations. We also convey this direction in Discussion in line 465 where we mention future research should incorporate “dynamic pH sensitivity”.

Reviewer #3:

Our Overall Response:

We appreciate the reviewer's constructive feedback. We believe the insightful comments have significantly improved the quality of our manuscript. Shown below are our point-by-point responses. Any new text added to the revised manuscript is shown in red.

Reviewer #3 Point #1

The authors should explicitly discuss their specific methods for data harmonization and standardization across the diverse experimental conditions included in LNPDB. Given the dataset encompasses varying experimental parameters such as distinct cell lines, diverse target organs, lipid compositions and molar ratios, differing lipid-to-nucleic acid ratios, dosage variations, and varying reporting formats for luminescence measurements (including units and normalization methods), a detailed explanation of how these factors are systematically standardized or normalized is necessary. Clearly articulating this process will enhance understanding of the dataset's consistency, reliability, and suitability for predictive modeling and comparative analyses across different studies.

Our Response to Reviewer #3 Point #1

We thank the reviewer for this suggestion to further elaborate on our featurization methods. We agree further discussion on featurization is advantageous. We have revised our Methods subsections "Data collection" and "Featurization" to make these points clearer.

- To better clarify our literature screening strategy for the publications in LNPDB, we have added the following sentences in lines 483-486 of Methods.

"In summary, publications were selected from the literature based on the presence of large screening datasets, primarily focused on ionizable lipids, to allow for meaningful within-dataset comparisons. Additional publications were selected to broaden representation of helper lipids, cholesterol, and PEG lipids."

- Rather than simply citing a prior study (Witten et al., *Nat. Biotechnol.* (2024)) for details about how delivery results were extracted from papers, we now explicitly reiterate more details in the following sentences in lines 487-496.

"Functional data – most commonly delivery performance – were extracted from published heatmaps and bar plots by digitizing the figures and interpolating values based on either the color scale (heatmaps) or bar height (bar plots) as defined in the accompanying legends. Because delivery values are often reported on different scales across studies and modalities, for each publication and for each delivery context (e.g., in vivo or in vitro within the same publication), functional delivery data were standardized to have mean 0 and

standard deviation 1. When raw luminescence values spanned several orders of magnitude, they were first log-transformed prior to standardization. Thus, datapoints were treated as directly comparable within individual screens, but not necessarily across different screens or assay modalities, and transformations were applied to prevent overemphasis of any single dataset.”

With respect to improving cross-study comparisons, we have our comment in Discussion in lines 454-455 regarding encouraging “researchers to incorporate standardized LNP controls in future in vitro and in vivo screens to enable more effective cross-study comparisons.”

Reviewer #3 Point #2

The authors should refer to the article (<https://doi.org/10.1021/acscabm.4c01716>) as this study explicitly investigates the impact of various lipid compositions, including helper lipids, cholesterol, and PEG-lipids, on LNP performance. The manuscript should clearly delineate how LNPDB advances or expands upon such existing work.

Our Response to Reviewer #3 Point #2

We thank the reviewer for bringing this to our attention.

- First, we note that this paper was published after we prepared our draft, and we have since reviewed and cited this paper in the following sentence in Introduction in lines 92-94. “All of these methods, including a recent effort to synchronize data across studies¹⁸, reflect important first steps in bringing machine learning to lipid nanomedicine; however, there are areas for improvement.”
- As for how LNPDB expands upon this work, LNPDB features greater than 3x more LNPs (see our response to Reviewer #3 Point #3 for information regarding the data added to LNPDB for this revised manuscript). Moreover, LNPDB introduces a new modality – molecular dynamics – for featurization. Next, LNPDB offers a platform for researchers to upload *new* LNP structure-function data, enabling LNPDB to expand over time as new data are added. Lastly, our manuscript introduces our new state-of-the-art deep learning model LiON. We have revised the manuscript to make this point clearer in lines 254-256 and lines 111-116.

“Importantly, LNPDB establishes a framework of training data for the continued development of next-generation deep learning models for LNP design.”

“Future user contributions are supported, enabling LNPDB to expand over time as new data are deposited. Additionally, LNPDB provides CHARMM²⁴ force field topology and

parameter files for all constituent lipids, allowing molecular dynamics (MD) simulations to generate three-dimensional, time-resolved lipid data that can enhance predictive modeling. For rational LNP design, MD simulations offer a new modality to generate dynamic structural data for lipids not readily accessible with current experimental methods.”

Reviewer #3 Point #3

The dataset described in the manuscript (as shown in Fig. 2b-c) exhibits significant imbalance. Specifically, despite featuring over 10,000 distinct ionizable lipids, the distribution is heavily skewed: most ionizable lipids contain only up to four nitrogen atoms (despite a range of 1–17 nitrogen atoms), approximately 50% of ionizable lipids have tail lengths concentrated at 18 or 19 carbons (despite a broader range of 3–37 carbons), and most formulations use an ionizable lipid-to-nucleic acid ratio of about 10. Additionally, only seven helper lipid types are represented, with nearly half (45.7%) being DOPE, followed by DSPC (26.9%) and DOTAP (14.4%), leaving minimal representation (<5%) for other types such as MDOA, DDAB, 14:0 PA, and 18:0 PG. The dataset is similarly limited for PEG-lipids, predominantly including only DMG-PEG2000 and DMPE-PEG2000. Given these substantial imbalances, the authors need to thoroughly discuss potential impacts on predictive accuracy, model generalizability, and potential biases introduced by such imbalances. They should explicitly address how these limitations might influence the reliability of data-driven models trained on this dataset and provide a justification or rationale for how the current dataset can still effectively support accurate and generalizable modeling efforts.

Our Response to Reviewer #3 Point #3

We thank the reviewer for this perspective.

- First, we have significantly expanded LNPDB to better capture the LNP design space, notably expanding diversity in the components besides the ionizable lipid. This has led to marked increases in the overall quantity and diversity of LNPs in LNPDB and thus reduces some issues of imbalance. For example, the dataset now features more cholesterol types and PEG types (so no longer just DMG-PEG2000 and DMPE-PEG2000).

Altogether, the revised database has increased...

- o from 17,116 to 19,528 LNPs
- o from 10,524 to 12,845 unique ionizable lipids, and
- o from 28 to 42 publications

Because new datasets have been added, LNPDB now has more rows (i.e., LNP formulations). Additionally, we have added more columns (i.e., features) to LNPDB to better represent each LNP. These new columns include *Aqueous_buffer*, *Dialysis_buffer*,

fifthcomponent_name, *fifthcomponent_SMILES*, *fifthcomponent_molratio*,
Dose_ug_nucleicacid.

All figures are thus updated to reflect this new larger version of LNPDB.

- Here is a more detailed breakdown of the 14 specific publications added to LNPDB. Note that this information is also reflected in the updated Fig. 2, Table S1, and Fig. S1.
 - Study evaluating from Moderna evaluating SM-102 and 9 other ionizable lipids. This study was added due to the clinical significance of SM-102 for the field.
 - SS_2018 (<https://doi.org/10.1016/j.ymthe.2018.03.010>)
 - Study evaluating different cholesterols. This study was added to increase chemical diversity of cholesterols.
 - SP_2020 (<https://www.nature.com/articles/s41467-020-14527-2>)
 - Studies evaluating different PEG lipids. These studies were added to increase chemical diversity of PEG lipids.
 - LL_2025 (<https://doi.org/10.1039/D5NR00433K>)
 - LX_2025 (<https://doi.org/10.1016/j.jconrel.2025.01.071>)
 - Studies evaluating different fifth component LNP formulations. These studies were added to introduce fifth-component design LNPs into LNPDB.
 - SB_2024 (<https://doi.org/10.1016/j.bioactmat.2024.05.012>)
 - SY_2025 (<https://doi.org/10.1002/adhm.202403366>):
 - Studies evaluating new ionizable lipid libraries. These recent studies were added to further increase ionizable lipid diversity.
 - JL_2024
(<https://jnanobiotechnology.biomedcentral.com/articles/10.1186/s12951-024-02919-1>)
 - LX_2024_2 (<https://pubs.acs.org/doi/10.1021/jacs.4c10265>)
 - LX_2024_3 (<https://www.nature.com/articles/s41565-024-01747-6>)
 - SW_2024
(<https://www.sciencedirect.com/science/article/abs/pii/S0168365924007016>)
 - XH_2024_2 (<https://www.nature.com/articles/s41551-024-01267-7>)
 - AP_2025 (<https://www.nature.com/articles/s42004-025-01516-z>)
 - SX_2025 (<https://www.nature.com/articles/s41551-025-01480-y>)
 - XH_2025 (<https://www.biorxiv.org/content/10.1101/2025.02.25.640222v1>)
- Nevertheless, some issues of imbalance (e.g., overrepresentation of DOPE/DSPC/DOTAP, disproportionate representation of a specific ionizable lipid-to-nucleic acid ratio of 10) still

remain, as this reflects the historical bias of the field toward certain lipid compositions and ratios. We agree that further discussion is warranted to explain how these imbalances may influence data-driven models. We have added/expanded on the following sentences in lines 436-449 in Discussion.

“Furthermore, MD is data-efficient: it does not require large training datasets, making it beneficial for evaluating novel chemistries as well as underrepresented formulation spaces that existing datasets do not effectively capture. This is especially relevant for LNPs, where the large combinatorial space of lipid types and ratios results in sparse data that can pose a major challenge for machine learning model generalization. For example, when screening novel or underrepresented helper lipids, ML models trained on LNPDB may struggle to generalize because the dataset is disproportionately enriched for DOPE, DSPC, and DOTAP (**Fig. 2c**), reflecting the field’s longstanding reliance on these lipids. In such cases, MD can provide complementary value by directly modeling the physical interactions of these underrepresented helper lipids, offering mechanistic insights that are not dependent on prior training data. Moreover, MD is also generally well-suited for small molecules like lipids, where dynamic shape, orientation, and local interactions can have outsized effects on function. In future applications, MD and machine learning may complement one another, with MD simulations contributing dynamic structural data as input features for deep learning models.”

- Lastly, the data imbalance concern is a good observation which warrants further discussion of how this concern manifests depending on the specific LNP library in question. To address this question, we have added the following sentences in lines 254-259 in Results. “Importantly, LNPDB establishes a framework of training data for the continued development of next-generation deep learning models for LNP design. Moreover, given our results demonstrate significant variation in model accuracy across libraries, future research can leverage the training data of LNPDB to design alternative models that may be better suited for the specific LNP design strategy (e.g., helper lipid optimization) or delivery target (e.g., in vivo muscle) under investigation.”

Reviewer #3 Point #4

Although the authors have visually demonstrated diversity using UMAP embeddings (Fig. 2a), the overlaps visible between data points from different sources suggest that additional quantitative measures, such as Tanimoto similarity scores or cluster purity metrics, would be beneficial. Providing

such quantitative analyses would strengthen the authors' claims about the distinctiveness of LNP formulations sourced from different studies.

Our Response to Reviewer #3 Point #4

We thank the reviewer for this suggestion.

- We have added Fig. S2 (shown below). In Fig. S2a, as the reviewer suggested, we demonstrate how LNP nearest neighbors *within a single library* have higher similarity than those *across different libraries*. Note we use cosine instead of Tanimoto to compute similarity, as the latter is ideal for binary values but our feature space is not entirely binary. Additionally, in Fig S2b, we thought it would be interesting to consider an alternative UMAP embedding approach. Here, we extract the encodings from the penultimate linear layer of the LiON model's feedforward neural network trained on LNPDB; we then perform UMAP dimensionality reduction on these encodings ("fingerprints").

Fig. S2: Additional LNP embedding analysis. **a** Probability distributions of cosine similarities for the top 50 nearest neighbors for each LNP, computed separately within library and across libraries. This analysis is based on the original LNP embedding space shown in Fig. 2a prior to UMAP dimensionality reduction. Probability density functions are plotted, with vertical lines indicating median values and significant Kolmogorov-Smirnov p value shown (Kolmogorov-Smirnov statistic = 0.76). Intra-library LNP neighbors show higher similarity, reflecting greater homogeneity within studies, whereas inter-experiment LNP neighbors are more broadly distributed. **b** UMAP visualization of high-dimensional LNP embedding landscape (i.e., "fingerprints") from LiON model of LNP formulations. Fingerprints were extracted from the penultimate linear layer of the LiON model's feedforward neural network trained on LNPDB as shown in Fig. 3a. This is an alternative embedding approach to that visualized in Fig. 2a. Source data are provided as a Source data file.

- Moreover, in support of this new Fig. S2, we have added the following sentences in lines 173-176 in Results.

“This is reinforced by our finding that within-library LNP pairs exhibit significantly higher cosine similarity than across-library pairs (**Fig. S2a**), and a UMAP of LiON-learned LNP fingerprints similarly yields library-specific clusters, albeit with less pronounced separation (**Fig. S2b**).”

Reviewer #3 Point #5

The authors have employed Spearman correlation coefficient as their primary evaluation metric (Fig. 3a-b). Given that Spearman correlation assesses rank correlation rather than absolute predictive accuracy, it might not fully capture predictive performance, particularly in practical applications where precise quantitative predictions are essential. The authors should consider or discuss the applicability and advantages of additional metrics such as Pearson correlation, mean absolute error (MAE), root mean squared error (RMSE), or even classification-based metrics if delivery outcomes could be discretized. Justifying the choice of Spearman correlation and/or demonstrating the robustness of their model across these alternative metrics would significantly enhance the manuscript.

Our Response to Reviewer #3 Point #5

We thank the reviewer for this suggestion.

- For our machine learning results in Fig. 3, we opted for rank-based Spearman correlations for a couple reasons. First, Spearman does not assume normality, and some publications have somewhat skewed experiment values (see Fig. S1b bottom). Second, rank-based correlations can perform better at the margins for non-normal distributions, and, during the design and testing of an LNP library, a practical consideration for an in silico method is whether it can identify the *best* (rank #1) formulation out of some potential library of many candidate LNPs. For these reasons, we have opted to maintain Spearman correlations for our main Fig. 3. However, we agree that additional analysis using linear Pearson correlation is useful with which to compare. Accordingly, we have added Fig. S3 (shown below), which presents the same analysis as the main Fig. 3 except uses Pearson correlations instead of Spearman correlations. The same key results are largely replicated, i.e., LiON achieves improved predictive performance for 5 out of the 7 test datasets when trained on the larger LNPDB dataset, and LiON trained on LNPDB achieves better predictive performance compared to AGILE for the 4 held-out test sets evaluated. Note that SL_2020 has the smallest sample size and thus shows more divergence between Spearman and Pearson results.

Fig. S3: Pearson correlation results for deep learning models. a Analogous to Fig. 3a but measured using Pearson correlation. **b** Analogous to Fig. 3b but measured using Pearson correlation. Source data are provided as a Source data file.

- This suggestion to emphasize linear correlation strategies where possible has motivated us to change our correlation method for reporting molecular dynamics results in Figs. 4 (shown below), S8, S9, S11. The underlying distributions for these molecular dynamics LNPs fit the normality criterion well and thus are good candidates for the linear Pearson correlation method. Hence, we changed the reported r and p values for our molecular dynamics results from Spearman to Pearson.

Fig. 4: LNPDB facilitates MD simulations of LNP membrane dynamics, uncovering new structure-function relationships towards predicting delivery performance. a Schematic illustration of the LNP MD simulation workflow enabled by LNPDB: select an LNP formulation of interest, initialize a representative bilayer with the provided CHARMM

force field topology and parameter files for the constituent lipids in the selected LNP formulation, run an MD simulation, and extract features from the resulting trajectory to assess correlation with experimental delivery performance. **b** Simulated bilayers for three example LNP formulations available in LNPDB: LNP_0009476 (left; Onpattro formulation from ZH_2023⁴³), LNP_0003881 (center; from LM_2019⁷), and LNP_0004075 (right; from LM_2019⁷). Snapshots are taken at 1.5 μ s for LNP_0009476 and LNP_0003881, and 100 ns for LNP_0004075. For LNP_0009476 and LNP_0003881, which formed stable bilayers, separate simulation snapshots are shown for the fully-neutral and half-protonated ionizable lipid (IL) conditions; density profiles are provided. For LNP_0004065, which failed to form a stable bilayer as indicated by the escaped ionizable lipids, the simulation snapshot from the half-protonated ionizable lipid condition is shown. **c** Violin plot demonstrating that simulated bilayer stability of $N = 54$ LNP formulations from LM_2019⁷ is predictive of experimental delivery performance, as measured in the LM_2019 study by normalized luciferase activity in HeLa cells. p value resulting from two-sided Wilcoxon signed-rank test is shown. **d** Method for calculating critical packing parameter (CPP). Analyses are performed on every 1 ns from the final 500 ns of 1.5 μ s trajectories of LNP formulations that formed stable, equilibrated bilayers. For each of these timesteps, the coordinates for ionizable lipids are extracted, and the CPP value is computed for each ionizable lipid. **e** For $N = 34$ LNP formulations from LM_2019⁷ that formed stable bilayers (subset from original $N = 54$ as $N = 20$ were unstable), the mean CPP_V value – computed as $CPP_V = V/(a_{olc})$ of ionizable lipids – significantly correlates with experimental delivery performance; this is the case for both protonated ionizable lipids in the half-protonated systems (top), as well as neutral ionizable lipids in the fully-neutral systems (bottom). Pearson r and p values are noted for the overall trend as well as for each ionizable lipid amine group (A12, A2, A3), which are also represented by point color. **f** Comparable analysis to Fig. 4e except an alternative method for computing CPP is used: $CPP_{RG} = R_g^{tail}/R_g^{head}$, where R_g denotes radius of gyration. The CPP_{RG} method yields similarly robust correlations with experimental delivery performance. Points denote mean values, and error bars denote +/- SEM for CPP_V and standard deviation for CPP_{RG} . Additional details of all bilayer simulations analyzed in this study are provided in Table S2. Source data are provided as a Source data file.

Reviewer #1:

*The authors have made substantial revision to the ms, and conducted additional analyses (e.g. CPP).
I am satisfied with the revisions and support the publication of this timely contribution.*

Our Response:

We thank the reviewer for recognizing the value of our work and supporting its publication.

Reviewer #2:

Our Overall Response:

We appreciate this feedback. Shown below are our point-by-point responses.

Reviewer #2 Point #1

This study presents methodological concerns regarding data integration. The authors employed simple log-transformation and standardization before directly merging data from different studies. However, such basic standardization cannot effectively eliminate systematic biases arising from different laboratories, experimental conditions, and measurement protocols. This approach may create a misleading impression of data comparability and potentially introduce issues in subsequent model training. More rigorous data integration methods should be employed to address data integration challenges.

Our Response to Reviewer #2 Point #1

We thank the reviewer for inviting additional discussion here.

- The matter of systematic biases from different labs/experiments is a fundamental limitation of the field itself, one whose effect we sought to minimize by transformation and standardization that we mentioned in Methods. Although it would be ideal for all compiled data to emerge from a single lab, limiting our current dataset in such a way would significantly reduce the quantity and diversity of LNPs. Our chosen standardization approach represented a practical strategy to enable cross-study comparison given the available data, and we acknowledge in lines 498-507 of Methods that this approach is imperfect.

“Predictive performance results were separated by publication, as datapoints were treated as directly comparable within individual screens, but not necessarily across different screens or assay modalities. Moreover, standardization was applied to prevent overemphasis of any single dataset. The deep learning models used in this study do train across datasets to maximize the diversity of trainable data, though we recognize that the ability of data from one study to inform structure-function relationships in another is limited by inherent differences in experimental protocols, measurement modalities, and assay sensitivities across studies that may introduce systematic biases. LNPDB introduces experimental condition variables (e.g., solvents, dose) towards bridging studies, but these additions can only partially mitigate the systematic differences across laboratories and experimental setups.”

- Moreover, this discussion reinforces why we argue in Discussion in lines 455-462 for the inclusion of standardized controls for the field going forward, so as to enable more effective cross-study comparisons.

“A key limitation of current LNP data – including those compiled in LNPDB – is the difficulty of comparing across studies due to variability in experimental methods (e.g., dose, cell type, animal model, nucleic acid purity, imaging equipment, injection technique, etc.). By establishing LNPDB as a centralized repository, we aim to encourage researchers to incorporate standardized LNP controls in future in vitro and in vivo screens to enable more effective cross-study comparisons. These standardized LNP controls could be Spikevax² (50% SM-102, 10% DSPC, 38.5% cholesterol, 1.5% DMG-PEG2000) or Onpattro⁷⁴ (50% DLin-MC3-DMA, 10% DSPC, 38.5% cholesterol, 1.5% DMG-PEG2000), FDA-approved LNPs for COVID-19 and transthyretin-induced amyloidosis, respectively.”

Reviewer #2 Point #2

The marked performance variations of the LiON model across different databases are likely attributable to inadequate data integration strategies, which may systematically obscure the true predictive value and distinctive contributions of the LNPDB database. Alternative modeling approaches should be considered, such as dataset-specific modeling or multi-task learning frameworks to more appropriately integrate heterogeneous data sources, thereby yielding more reasonable and reliable model results to reflect the LNPDB quality.

Our Response to Reviewer #2 Point #2

We thank the reviewer for inviting additional discussion and analysis here, similar to Point #1.

- Towards dataset-specific modeling, we introduce Fig. S5. We trained our LiON model on individual datasets (70%-15%-15% train-validation-test split) and compared performance to the same model trained on the integrated LNPDB dataset. As shown in Fig. S5, models trained exclusively on a single dataset (three randomly selected) yielded poorer predictive performance than those trained across studies, supporting the benefit of data integration across studies in capturing broader structure-function relationships, even in spite of the limitations of data integration that we noted in Point #1.

Fig. S5: Learning from outside datasets improves predictive performance of LiON. Training LiON exclusively on a single dataset yields poorer predictive performance (measured by Spearman correlation) than when trained on LNPDB. This suggests that integrating data from multiple studies enables the model to capture broader structure-function relationships. Source data are provided as a Source data file.

- Accordingly, we have added the following sentence in lines 244-248 in Results.
“Moreover, despite limitations of integrating data from multiple studies as discussed in Methods, LiON models trained across datasets achieved higher predictive performance than those trained on single datasets (**Fig. S5**), suggesting that training across multiple studies in LNPDB enabled LiON to learn more generalizable structure-function relationships (**Fig. S5**).”
- Lastly, we hope that introducing LNPDB as a public, centralized data repository will motivate future research to design other deep learning models. Our intention is not to present our LiON model shown here as a finalized approach, but rather as a foundation upon which future, more specialized models can be developed. This echoes our discussion in lines 258-263.
“Importantly, LNPDB establishes a framework of training data for the continued development of next-generation deep learning models for LNP design. Moreover, given our results demonstrate significant variation in model accuracy across libraries, future research can leverage the training data of LNPDB to design alternative models that may be better

suited for the specific LNP design strategy (e.g., helper lipid optimization) or delivery target (e.g., in vivo muscle) under investigation.”

Reviewer #2 Point #3

The authors used two algorithms to calculate the CPP values of 34 different LNP formulations from the LM_2019 library (lines 340–365) and subsequently correlated them with experimental delivery performance (Fig. 4). The current sample size (34) is too small to represent the 19,528 LNPs and 12,845 unique ionizable lipids in the LNPDB. It is recommended to increase the number of tested formulations to further validate the correlation and conclusion obtained in this study.

Our Response to Reviewer #2 Point #3

- This manuscript involved simulating a total of 134 bilayers (77 fully-neutral, 57 half-protonated) from the LM_2019 dataset. As we state in lines 646-648 of Methods, “**This subset, drawn from a single combinatorial ionizable lipid library, was chosen as a representative example of systematic lipid library design commonly employed in the field, while keeping the scope feasible within computational limits.**” The reviewer is correct that after selecting for only the *stable* bilayers, the CPP analysis shown in Fig. 4e-f includes 34 simulated bilayers each for protonated and neutral conditions. Note that the preceding stability analysis in Fig. 4c includes 54 simulated bilayers.
- The central focus of this manuscript is developing LNPDB and demonstrating two proof-of-concept applications for deep learning and molecular dynamics simulations. We do not claim that the subset of LNPs that we simulated in this study represent all ~20,000 LNPs in the entire database. This is stated in lines 301-308 of Results.
“**To demonstrate how LNPDB can be used to facilitate MD simulations, we simulated the bilayer equilibration process for a subset of LNP formulations and extracted structural features to assess correlation with experimental transfection (Fig. 4a). This use case represents just one of many potential simulation strategies enabled by the database (see “Discussion”).**”
- Moreover, as we state in lines 464-475 of Discussion, we encourage future research to use the topology and parameter files provided in LNPDB to expand simulation efforts beyond the dataset evaluated in this study.
“**Future research should explore experimental validation of MD-derived features such as CPP. The MD bilayer models presented in this work provide a simplified yet informative framework that yields significant correlations with delivery performance for the evaluated dataset. However, future research should leverage the topology and parameter files**

provided in LNPDB to expand simulation efforts to include additional delivery-relevant phenomena, such as membrane fusion dynamics, interactions with nucleic acids, and dynamic pH sensitivity during endosomal escape. To support MD simulations with larger system sizes, longer time scales, and the inclusion of nucleic acids, we plan to incorporate Martini 3 coarse-grained lipid and nucleic acid parameters⁷⁵ in future versions of LNPDB. This will facilitate efficient simulations for many more LNPs to further explore structure-function relationships. Moreover, although this version of LNPDB includes some LNPs with five lipid components, future versions will further incorporate LNPs with more than four components (e.g., additional lipids⁸ or lipids conjugated to targeting ligand⁷⁶).

- We are actively working on simulating many more LNP systems in different contexts, as alluded to in the above sentences in Discussion. These will be included in future publications. For the sake of this manuscript whose principal aim is to introduce LNPDB, we consider our current simulation data to be sufficiently robust.

Reviewer #2 Point #4

We have noticed that this study currently considers different quantities of auxiliary lipids (CHOL, DSPC, DOPE) when calculating CPP, with DOPE being the main one. Therefore, whether the differences and changes in the quantity of these helper lipids have an impact on the ionizable lipid CPP should be further compared and explained. The revised manuscript only mentions that the impact of PEG is not significant.

Our Response to Reviewer #2 Point #4

- The molecular dynamics analysis in this manuscript focused on the LM_2019 dataset, which varied the ionizable lipid type. The reviewer makes a great point about exploring how other helper lipid types may affect the CPP of the ionizable lipid. We are actively working on simulating different LNPs with different helper lipids and ratios and thus far have promising preliminary results. These results will be reported in a future publication where novel LNP formulations will be disclosed. Echoing our response to Point #3, the main focus of this manuscript is to introduce LNPDB.
- As requested, we introduced the PEG lipid analysis in response to this Reviewer #2's Point #6 during the first revision round to demonstrate its inclusion has no significant effect on CPP.

Reviewer #2 Point #5

The authors noted that when only considering the subset with CPP > 1 (Fig. 4 & Fig. S9), the correlative performance improved substantially for both the CPPV and CPPRg methods. Since the subset size was significantly reduced to fewer than 10, the improvement in correlation is expected. However, it remains uncertain whether this conclusion would hold when more ionizable lipids are included.

Our Response to Reviewer #2 Point #5

- Subsetting by CPP > 1 did not lead to the overall sample size being less than 10. As shown in the pre-existing Fig. S10 (formerly titled Fig. S9), the overall subset sample size ranges from 16 to 27 (see red text in figure) depending on the protonation condition being considered. These sample sizes are sufficiently robust for correlative analysis. The overall correlation coefficient values (see red text in figure) are indeed higher than their non-subsetted counterparts as shown in the main Fig 4e-f. Thus, we hold that our main claim regarding improved predictive performance is substantiated.
- The reviewer is correct that the subsets *once stratified by amine* are usually less than 10 depending on protonation condition, which could limit conclusions. To no longer highlight any correlation results for sample sizes less than 10, we now have omitted stating the correlation coefficient values of the amine-specific analysis in lines 356-362.

“Moreover, when we focused our analyses on the subset of LNPs with mean CPP values greater than 1 – corresponding theoretically to a transition to negative curvature – the correlative performance improved substantially for both the CPP_v method (protonated: overall r = 0.723; neutral: overall r = 0.680) and CPP_{Rg} method (protonated: overall r = 0.621; neutral: overall r = 0.646) (Fig. S10). Some correlations for amine 3 did not reach statistical significance, likely due to its limited representation of LNPs with CPP > 1.”

Reviewer #2 Point #6

The authors mentioned that using the CPPV method (lines 670-674), "...CPPV values exhibit greater variability across lipids and time steps compared to CPPRg...". Such variability may be normal. It is suggested to perform a cluster analysis to relate the complexity of LNP formulations and the polymorphism of ionizable lipids to delivery efficiency.

Our Response to Reviewer #2 Point #6

We thank the reviewer for inviting additional analysis here to assess whether the *variance of CPP* may itself be a predictor of delivery efficiency.

- Towards this, we introduce Fig. S13, shown below. We assess whether the standard error of the mean for CPP_V (Fig. S13a) or standard deviation for CPP_{Rg} (Fig. S13b) normalized by the mean CPP value correlates with delivery performance. Interestingly, to the reviewer's point, this variance measure for CPP_V significantly correlates with delivery performance. This significant relationship suggests greater ionizable lipid polymorphism (as measured by greater CPP_V variance) allows for more effective delivery, potentially due to increased capacity to accommodate more inverse-conical lipid geometries. This relationship, however, is not found to be significant for the CPP_{Rg} metric.

Fig. S13. Relationship between CPP variance and delivery performance. **a** Analogous to Fig. 4e but assessing SEM normalized by mean CPP_V values. This significant relationship suggests greater ionizable lipid polymorphism (as measured by greater CPP_V variance) allows for more effective delivery, potentially due to increased capacity to accommodate more inverse-conical lipid geometries. **b** This relationship is insignificant for CPP_{Rg} . SEM denotes standard of the mean. SD denotes standard deviation. Pearson r and p values are noted. Source data are provided as a Source data file.

- Accordingly, we have added the following sentence in lines 372-375 in Results.
“Interestingly, CPP_V variance also significantly predicts delivery performance, suggesting that greater ionizable lipid polymorphism allows for more effective delivery, potentially due to increased capacity to accommodate more inverse-conical lipid geometries (**Fig. S13**).”

Reviewer #2 Point #7

A comparison of computational equipment and time consumption between the CPPV and CPPRg methods should be provided to help other researchers choose the appropriate method.

Our Response to Reviewer #2 Point #7

We thank the reviewer for this point.

- In lines 589-595 of Methods, we now state the following.

“We ran all MD simulations on 48 NVIDIA RTX A5000 GPUs in parallel. Across the neutral systems, the average throughput was 363.86 ± 14.93 ns/day; across the protonated systems, it was 319.11 ± 11.65 ns/day. For the LM_2019 bilayers with run duration of 1.5 μ s, this corresponds to 4.1 days per neutral system and 4.7 days per protonated system. The full LM_2019 simulation batch, run on 48 GPUs (one per simulation), completed in about 10 days. Subsequent CPP calculations for all systems, executed on 384 CPU cores in parallel, finished in 5 hours, with comparable compute times for CPP_V and CPP_{Rg}.”

Reviewer #3:

I appreciate the authors' careful and thorough revision. In my view, all of my prior concerns have been adequately addressed in both the main manuscript and the Supporting Information. The additions and clarifications substantially improve the clarity, rigor, and reproducibility of the work. I therefore recommend the manuscript for publication. By way of summary: the revised version clearly documents data processing and standardization, positions the contribution relative to prior efforts, expands and analyzes the dataset to contextualize limitations, and provides quantitative checks that support the principal conclusions. These updates resolve my earlier requests and strengthen the manuscript's impact and transparency.

Our Response:

We thank the reviewer for their constructive comments during the first round and endorsement of the revised manuscript.